# Comparing the information content of probabilistic representation spaces

**Kieran A. Murphy** [ID]
*Dept. of Bioengineering, University of Pennsylvania*

*kieranm@seas.upenn.edu*

**Sam Dillavou** [ID]
*Dept. of Physics & Astronomy, University of Pennsylvania*

*dillavou@sas.upenn.edu*

**Dani S. Bassett** [ID]

*dsb@seas.upenn.edu*

*Depts. of Bioengineering, Electrical & Systems Engineering, Physics & Astronomy, Neurology, Psychiatry, University of Pennsylvania; The Santa Fe Institute; The Neuro, Montreal Neurological Institute, McGill University*

**Reviewed on OpenReview:** *https://openreview.net/forum?id=adhsMqURI1*

## Abstract

Probabilistic representation spaces convey information about a dataset and are shaped by factors such as the training data, network architecture, and loss function. Comparing the information content of such spaces is crucial for understanding the learning process, yet most existing methods assume point-based representations, neglecting the distributional nature of probabilistic spaces. To address this gap, we propose two information-theoretic measures to compare general probabilistic representation spaces by extending classic methods to compare the information content of hard clustering assignments. Additionally, we introduce a lightweight method of estimation that is based on fingerprinting a representation space with a sample of the dataset, designed for scenarios where the communicated information is limited to a few bits. We demonstrate the utility of these measures in three case studies. First, in the context of unsupervised disentanglement, we identify recurring information fragments within individual latent dimensions of VAE and InfoGAN ensembles. Second, we compare the full latent spaces of models and reveal consistent information content across datasets and methods, despite variability during training. Finally, we leverage the differentiability of our measures to perform model fusion, synthesizing the information content of weak learners into a single, coherent representation. Across these applications, the direct comparison of information content offers a natural basis for characterizing the processing of information.

## 1 Introduction

The comparison of representation spaces is a problem that has received much attention, particularly as a route to a deeper understanding of information processing systems (Klabunde et al., 2023; Mao et al., 2024; Huh et al., 2024; Lin & Kriegeskorte, 2024) and with applications in tasks like transfer learning, ensembling, and other forms of representational alignment (Sucholutsky et al., 2023; Muttenthaler et al., 2024). Existing methods are applicable to point-based representation spaces, including centered kernel alignment (CKA) (Kornblith et al., 2019) and representational similarity analysis (RSA) (Kriegeskorte et al., 2008). For representation spaces whose citizens are probability distributions, such as in variational autoencoders (VAEs) or in biological systems with inherent stochasticity, failure to account for the distributed nature of representations can miss important aspects of the relational structure between data points (Duong et al., 2023). Few methods account for the distributional nature of representations (Klabunde et al., 2023).

One of the few existing methods for comparing probabilistic representation spaces, stochastic shape metrics (Duong et al., 2023), relies on geometric assumptions and fixed transformations, which can limit its flexibility. In contrast, we adopt an information-theoretic approach that evaluates representation spaces based on the information they transmit, providing a method that is agnostic to properties like dimensionality or

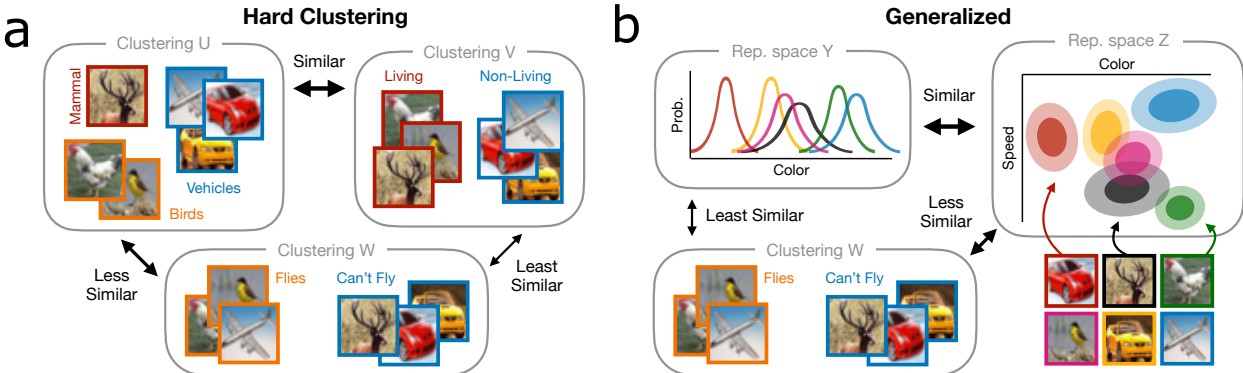

Figure 1: **Similarity of representation spaces.** In this work, we generalize measures to compare the information content of clustering assignments to apply to probabilistic representation spaces. **(a)** A hard clustering assignment, such as the living/non-living distinction conveyed by clustering $V$, communicates certain information about the dataset (here, CIFAR-10 images). Comparing the information content of different clustering assignments enables comparative analyses between algorithms, model fusion, and benchmarking. **(b)** We generalize measures for comparing hard clustering assignments to be applicable to probabilistic representation spaces, by recognizing the latter as soft clustering assignments. When cast in terms of information content, there is no requirement for the dimensionality of the spaces to match, and hard clusterings (e.g. labels or annotations) can be compared to probabilistic spaces.

whether the spaces are discrete or continuous. Specifically, we compare two probabilistic representation spaces using quantities derived from the mutual information between them, capturing their relational structure while ensuring broad applicability across diverse contexts.

We take as a motivating example the task of unsupervised disentanglement, whose goal is to break information about a dataset into useful factors of variation (Higgins et al., 2018; Locatello et al., 2019; Träuble et al., 2021; Balabin et al., 2023; Van Steenkiste et al., 2019). As an example, a representation space might be trained on images of cars so that color, orientation, and model information are separated into different latent dimensions without any supervision about such factors. When ground truth factors of variation are unavailable for evaluation, as is generally the case for real-world datasets, existing evaluation methods assess the degree of consensus in an ensemble of trained models (Duan et al., 2020; Lin et al., 2020). However, the relatedness of representation spaces has failed to account for the probabilistic nature of the representations, reducing posterior distributions to their means and then using point-based comparisons such as correlation (Duan et al., 2020). With a direct comparison of the information content of representation spaces, we stand to improve the characterization of consensus and of unsupervised disentanglement more generally.

In this work, we generalize measures of similarity of the information content of hard clustering assignments (Fig. 1a) to be applicable to probabilistic representation spaces (Fig. 1b). To motivate the generalization, we treat probabilistic representation spaces as soft clustering assignments of data, whereby partial distinguishability between data points is expressed by the overlap between posterior distributions. We then assess the information content of learned representation spaces produced by models in an ensemble of random initializations, and study the effects of method, training progress, and dataset on information processing.

## 2 Related work

The proposed method builds upon classic means of comparing the information content of different cluster assignments (clusterings) of a dataset and generalizes them to compare probabilistic representation spaces. We then use the method to empirically study generative models designed to fragment information about a dataset in a learned latent space, i.e., for unsupervised disentanglement.

**Similarity of clusterings and of representation spaces.** The capacity to compare transformations of data produced by different machine learning models enables ensemble learning, a deeper understanding of

methodology, and benchmarking (Punera & Ghosh, 2007). Strehl & Ghosh (2002) used outputs of clustering algorithms to perform ensemble learning, based on a measure of similarity between clustering assignments that we will extend in this work: the normalized mutual information (NMI). Referred to as consensus clustering or ensemble clustering, efforts to combine multiple clustering outputs can leverage any of a variety of similarity measures (Wagner & Wagner, 2007; Vinh et al., 2009; 2010; Vega-Pons & Ruiz-Shulcloper, 2011; Huang et al., 2017). Another measure backed by information theory is the variation of information (VI), coined by Meilă (2003) for clustering but recognized as a metric distance between information sources at least twice before (Shannon, 1953; Crutchfield, 1990).

Commonly referred to simply as 'clustering', *hard* clusterings assign every input datum to one and only one output cluster and have been generalized to multiple forms of *soft* clustering (Campagner et al., 2023). We focus specifically on *fuzzy* clustering, where membership is assigned to multiple clusters and must sum to one for each datum (Zadeh, 1965; Dunn, 1973; Ruspini et al., 2019). We observe that a probabilistic representation space communicates for each datum a soft assignment over latent vectors, with the degree of membership expressed by the probability density of posterior distributions. Comparisons of hard clustering have been extended to fuzzy clustering for measures that indirectly assess information content, such as the Rand index (Punera & Ghosh, 2007; Hullermeier et al., 2011; D'Ambrosio et al., 2021; Andrews et al., 2022; Wang et al., 2022). Information-based measures have been extended for specific types of soft clusters (Campagner & Ciucci, 2019; Campagner et al., 2023) but none, to our awareness, for comparing the information content of fuzzy clusterings over a continuous representation space.

A rich area of research compares point-based representation spaces via the pairwise geometric similarity of a common set of data points in the space (Kornblith et al., 2019; Hermann & Lampinen, 2020; Klabunde et al., 2023), building upon representational similarity analysis from neuroscience (Kriegeskorte et al., 2008). In place of geometric similarity, topological similarity more closely probes the information available for downstream processing by employing specific similarity functions with tunable parameters (Lin & Kriegeskorte, 2024; Williams, 2024). For comparing probabilistic representation spaces, stochastic shape metrics (Duong et al., 2023) extend a distance metric between point-based representation spaces based on aligning one with another through prescribed transformations (e.g., rotations) (Williams et al., 2021). By contrast, the information-theoretic lens we adopt requires no enumeration of transformations nor tuning of parameters, and directly assesses the information available for processing by downstream neural networks.

**Unsupervised disentanglement.** Disentanglement is the problem of splitting information into useful pieces, possibly for interpretability, compositionality, or stronger representations for downstream tasks. Shown to be impossible in the fully unsupervised case (Locatello et al., 2019; Khemakhem et al., 2020), research has moved to investigate how to utilize weak supervision (Khemakhem et al., 2020; Sanchez et al., 2020; Vowels et al., 2020; Murphy et al., 2022) and incorporate inductive biases (Balabin et al., 2023; Chen et al., 2018; Rolinek et al., 2019; Zietlow et al., 2021; Hsu et al., 2024).

A significant challenge for disentanglement is evaluation when no ground truth factorization is available. While one proposed route to evaluation relies on a characterization of the posterior distributions in a model (PIPE, Estermann & Wattenhofer (2023)), a more common approach assesses consensus among trained models in an ensemble of randomly initialized repeats. The motivation, first proposed in Duan et al. (2020), is that disentangled models are more similar to each other than are entangled ones because there are intuitively more ways to entangle information than to disentangle it. There, the similarity between two models was evaluated with an *ad hoc* function of dimension-wise similarity, which was computed as the rank correlation or the weights of a linear model between embeddings in different one-dimensional spaces. ModelCentrality (Lin et al., 2020) quantified similarity between models with the FactorVAE score (Kim & Mnih, 2018) where one model's embeddings serve as the labels for another model. However, none of these methods fully account for the distributional nature of representations, instead relying on point estimates such as posterior means or single embeddings. In contrast, our approach focuses on capturing the probabilistic structure of representation spaces, and additionally shifts the emphasis from model similarity to channel (or latent subspace) similarity under the premise that the fragmentation of information, central to disentanglement, is more naturally studied via the information fragments themselves.

## 3 Method

Our goal is to compare the information transmitted about a dataset by different representation spaces, and we will use the comparison of hard clustering assignments (e.g., the output of $k$-means) as a point of reference. Analogously to hard clustering (Fig. 1a), probabilistic representation spaces communicate a *soft* assignment over embeddings that is expressed by a probability distribution in the space for each data point (Fig. 1b).

While the proposed method can be applied to any representation space with probabilistic embeddings, we will focus primarily on variational autoencoders (VAEs) (Kingma & Welling, 2014). Let the random variable $X$ represent a sample $x \sim p(x)$ from the dataset under study. $X$ is transformed by a stochastic encoder parameterized by a neural network to a variable $U = f(X, \epsilon)$, where $\epsilon$ is a source of stochasticity. We will make use of two fundamental quantities in information theory (Cover & Thomas, 1999), the entropy of a random variable $H(Z) = \mathbb{E}_{z \sim p(z)}[-\log p(z)]$ and the mutual information between two random variables $I(Y; Z) = H(Y) + H(Z) - H(Y, Z)$. The encoder maps each data point $x$ to a posterior distribution in the latent space, $p(u|x)$, after which a point $u \sim p(u|x)$ is sampled for downstream processing. Commonly, the posterior distributions are parameterized as normal distributions with diagonal covariance matrices, which will facilitate many of the involved measurements but is not required for the method.

### 3.1 Comparing representation spaces as soft clusterings

Consider a *hard* clustering of data as communicating certain information about the data (Fig. 1a). By observing the cluster assignment $U$ instead of a sample $X$ from the dataset, information $I(X; U)$ will have been conveyed. For a hard clustering, every data point is assigned unambiguously to a cluster—i.e., $H(U|X) = 0$—which makes the communicated information equal to the entropy of the clustering, $I(X; U) = H(U)$. We note that maximizing communicated information, such as by assigning each data point to its own cluster, does not yield a useful representation. The value of a representation lies in the balance between the information preserved and the irrelevant variation discarded, highlighting the importance of assessing its specific information content.

Given two hard clustering assignments $U$ and $V$ for the same data $X$, the mutual information $I(U; V)$ measures the amount of shared information content they express about the data. Previous works have found it useful to relate the mutual information to functions of the entropies $H(U)$ and $H(V)$. Strehl & Ghosh (2002) proposed the normalized mutual information (NMI) as the **ratio** of the **mutual information** and the **geometric mean** of the entropies,

$$\text{NMI}(U, V) = \frac{I(U; V)}{\sqrt{H(U)H(V)}}. \tag{1}$$

Meilă (2003) proposed the variation of information (VI), a metric distance between clusterings that is proportional to the **difference** of the **mutual information** and the **arithmetic mean** of the entropies,

$$\text{VI}(U, V) = -2\left(I(U; V) - \frac{H(U) + H(V)}{2}\right). \tag{2}$$

*Soft* clustering extends hard clustering to allow each datum to have partial membership in multiple clusters, allowing partial distinguishability between data points to be communicated (Zadeh, 1965; Ruspini et al., 2019) (Fig. 1b). While soft clustering is predominantly performed over a discrete set of clusters, here we view each point $u$ in a continuous latent space as a cluster, with the posterior distribution $p(u|x)$ assigning membership over the continuum.

The utility of NMI or VI over raw mutual information largely resides in the standardization of values. For two identical hard clustering assignments, $U$ and $U'$, $\text{NMI}(U, U') = 1$ and $\text{VI}(U, U') = 0$. In contrast, soft assignments include additional entropy stemming from uncertainty that is unrelated to the information communicated about the dataset, as is evident from the relation $H(U) = I(X; U) + H(U|X)$ with $H(U|X) > 0$. As a result, the entropy terms in Eqns. 1 and 2 that ground the mutual information $I(U; V)$ have components that disrupt the standardization of values. To address this issue, we propose replacing the entropy of a clustering assignment with the mutual information between two copies of that assignment: specifically, substituting $H(U)$ with $I(U; U')$, and similarly for $V$. For hard clustering, $H(U) = I(X; U) = I(U; U')$,

making these quantities interchangeable. Only the third quantity, $I(U; U')$, maintains the standardization of NMI = 1 and VI = 0 for identical soft clustering assignments. The generalized forms then become

$$\text{NMI}(U, V) = \frac{I(U; V)}{\sqrt{I(U; U')I(V; V')}}, \text{ and} \tag{3}$$

$$\text{VI}(U, V) = -2 \left( I(U; V) - \frac{I(U; U') + I(V; V')}{2} \right). \tag{4}$$

The generalization leaves NMI and VI unchanged for hard clustering, where $I(U; U') = H(U)$, and enables the measures to be applied to both soft and hard assignments. We note that the generalized VI is no longer a proper metric, as the triangle inequality is not guaranteed (Appx. D).

The conditional independence of clustering assignments given the data—i.e., $I(U; V|X) = 0$—allows each mutual information term in Eqns. 3 and 4 to be rewritten with regard to information communicated about the data. Namely, $I(U; V) = I(X; U) + I(X; V) - I(X; U, V)$, and $I(U; U') = 2I(X; U) - I(X; U, U')$. Because we have access to the posterior distributions, it is easier to estimate the mutual information between the data $X$ and a representation space (or a combination thereof) than it is to estimate the information between two representation spaces (Poole et al., 2019).

The extended NMI and VI benefit from the generality of information theory: the information content of two probabilistic representation spaces can be compared regardless of dimensionality or parameterization of posteriors, and a soft clustering can be compared to a hard clustering, whether from a quantized latent space or a discrete labelling process (Fig. 1b).

### 3.2   Routes to estimation

We propose two routes to estimating NMI and VI that offer a tradeoff between precision and speed; both leverage the known posterior distributions to calculate the information transmitted about the dataset by combinations of representation spaces, $I(X; \cdot)$. The first route is to compute $I(X; \cdot)$ with a straightforward Monte Carlo estimate using the aggregated posterior over the entire dataset of size $L$,

$$I(X; U) = \mathbb{E}_{x \sim p(x)} \mathbb{E}_{u \sim p(u|x)} \left[ \log \frac{p(u|x)}{\sum_i^L p(u|x_i)} \right]. \tag{5}$$

For the second route, we can use a measure of statistical similarity between posteriors $p(u|x_1)$ and $p(u|x_2)$ in a given representation space—the Bhattacharyya coefficient (Kailath, 1967), $\text{BC}(p, q) = \int_{\mathcal{Z}} \sqrt{p(z)q(z)} dz$—to quickly "fingerprint" the information content of spaces with the pairwise distinguishability of a sample of data points (Murphy & Bassett, 2023). The BC between two multivariate normal distributions can be efficiently computed in bulk via array operations. Once a matrix $\text{BC}_{ij} := \text{BC}(p(u|x_i), p(u|x_j))$ of the pairwise values for a random sample of $N$ data points is obtained, a lower bound for the information transmitted by the channel can be estimated with $I(X; U) \geq \frac{1}{N} \sum_i^N \log \frac{1}{N} \sum_j^N \text{BC}_{ij}$ (Kolchinsky & Tracey, 2017). For fast estimation of the information content of a representation space with respect to a ground truth generative factor, we can treat the labels as an effective hard clustering according to that factor, where BC values are either zero or one. Finally, the matrix $\text{BC}_{ij}$ for the combination of spaces $U_1$ and $U_2$ is their elementwise product (Appx. C). In other words, receiving both of the messages from $U_1$ and $U_2$ leads to a distinguishability between data points that is simply the product of the distinguishabilities under $U_1$ and $U_2$ separately. Together, the properties allow us to fingerprint each representation space through Bhattacharyya matrices, and then perform all subsequent analysis with only the matrices— i.e., without having to load the models into memory again.

### 3.3   Discovering consistently learned information fragments via OPTICS clustering of latent dimensions

In the context of disentanglement, we are interested in the similarity of information contained in individual dimensions across an ensemble of models. We compute the pairwise similarities between all dimensions of all models—for example, 10 dimensions each from 50 models in an ensemble for a $500 \times 500$ matrix of similarity values in Sec. 4.2—and then use density-based clustering to identify information content that is found consistently. OPTICS (Ankerst et al., 1999) is an algorithm that orders elements in a set according to

similarity and produces a "reachability" profile where natural groupings of elements are indicated through valleys. An OPTICS profile is analogous to a dendrogram produced by hierarchical clustering, but can be more comprehensible for large sets of points (Sander et al., 2003). The consistency of information fragments in the latent dimensions of an ensemble of representation spaces can then be visualized with the OPTICS reachability profile, the reordered pairwise similarity matrix—which will have block diagonal form if the fragments are consistent—and the information contained about ground truth generative factors, if available.

### 3.4  Model fusion

Consider a set of representation spaces found by an ensemble of weak learners. In the spirit of "knowledge reuse" (Strehl & Ghosh, 2002) originally applied to ensembles of hard clusters, we might obtain a superior representation space from the synthesis of the set. In contrast to other assessments of representation space similarity (Duan et al., 2020; Kim & Mnih, 2018; Duong et al., 2023), the proposed measures of similarity are composed of mutual information terms, which can be optimized with differentiable operations by a variety of means (Poole et al., 2019). We optimize a synthesis space by maximizing its average similarity with a set of reference spaces, performing gradient descent directly on the encoding of the data points used for the Bhattacharyya matrices. Instead of requiring many models to remain in memory during training of a new encoder, the Bhattacharyya matrices are computed for the ensemble once and then used for comparison during training, and training is fast because of the involved array operations.

## 4  Experiments

### 4.1  Comparison of related methods on synthetic spaces

We first evaluated the similarity of synthetic representation spaces using ours and related methods (Fig. 2). A dataset of 64 points was embedded to one- and two-dimensional representation spaces, each communicating different information about the dataset. To assess the information available for downstream processing, we trained a classification head for each of the nine latent spaces to predict the input $x$ from a sample of the corresponding posterior, $u \sim p(u|x)$. As shown in Fig. 2b, the classification head's average predictions, $p(\hat{x}_j|x_i) = \mathbb{E}_{u \sim p(u|x_i)}[p(\hat{x}_j|u)]$, reflected the overlap of posterior distributions in latent space by assigning non-zero probabilities to multiple outputs. The structure of these output probabilities closely matched the similarity patterns in the distinguishability matrices of panel **c**, supporting our use of the latter as fingerprints of the information in the latent space. Unlike the classification head's outputs, the distinguishability matrices were directly accessible from the latent space without additional network training.

We evaluated the pairwise similarity between the nine latent spaces of panel **a** using several methods (Fig. 2d). To assess the information content available for downstream processing, we compared the average predictions of the trained classification heads for each input $x_i$ through the Jensen-Shannon divergence, $\langle \mathrm{JSD}(p^\alpha(\hat{x}|x_i)||p^\beta(\hat{x}|x_i)) \rangle_i$. If classification heads for latent spaces $\alpha$ and $\beta$ produce similar output distributions, the latent spaces likely share similar information content. Next, CKA (Kornblith et al., 2019) can use different measures of representation similarity as its basis of comparison; we used the inner product between means of the posterior distributions for the representational similarity in linear CKA, an exponential function of the Euclidean distance for nonlinear CKA, and the Bhattacharyya coefficient between posteriors as a statistical basis of representational similarity. Nonlinear CKA requires selecting a distance parameter for each space: we used the arithmetic mean of the standard deviation of every representation. Next, we computed similarity according to stochastic shape metrics (Duong et al., 2023). Finally, we computed VI and NMI, both with a Monte Carlo approach and the significantly faster Bhattacharyya fingerprint-based approach.

Spearman's rank correlation of the similarity values between all pairs of synthetic spaces reveals multiple relationships between our proposed measures and the baselines. First, for this small dataset, the Bhattacharyya estimates of NMI and VI are highly consistent with the Monte Carlo alternatives while offering a speed up of $100\times$. Second, VI captures similarity much the same as when comparing the outputs of downstream classification heads (JSD) even though the former can be evaluated directly from the latent space. Third, NMI relates the latent spaces in much the same way as the Bhattacharyya variant of CKA. While this *ad hoc* variant of CKA lacks the information-theoretic underpinnings of NMI, it offers an easy-to-use alternative by simply replacing a point-based distance with a statistical distance between representations.

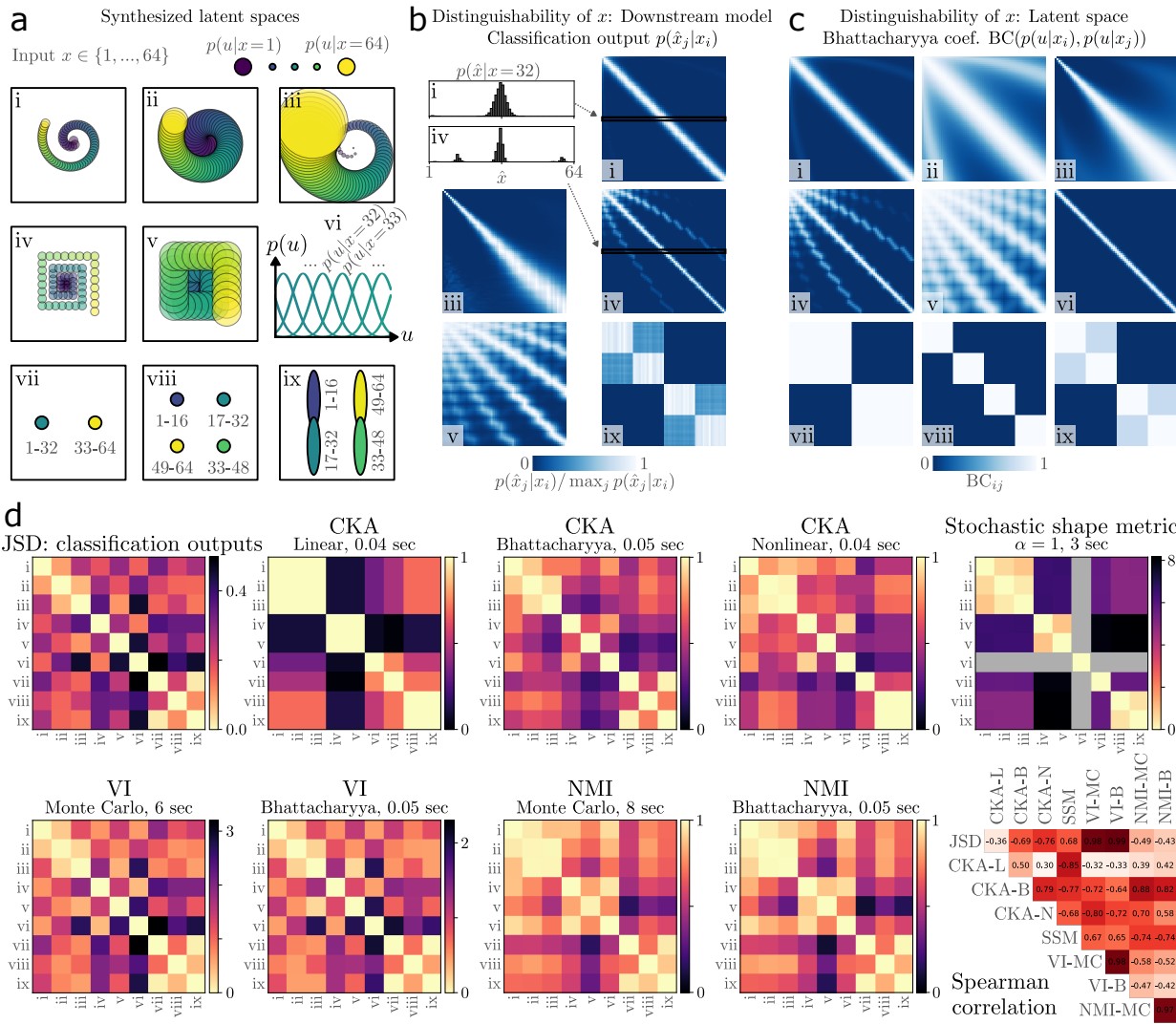

Figure 2: **Comparing similarity measures for synthetic embedding spaces.** **(a)** A dataset of 64 points, $x = 1, ..., 64$, is transformed into nine representation spaces marked **i-ix**. Each posterior distribution $p(u|x)$ is a Gaussian with diagonal covariance matrix and standard deviations indicated by the colored ellipses. **(b)** We trained a classification head on top of the latent spaces of **a** to predict the input $x$, given a sample from the posterior distribution $p(u|x)$. The predicted probability distributions $p(\hat{x}_j|x_i)$ are displayed as a matrix, with row corresponding to the input $x_i$. **(c)** The pairwise distinguishability of data points $x_i$ and $x_j$, as computed by the Bhattacharyya coefficient, serves as a fingerprint of the information content of the latent space. **(d)** The pairwise similarity of the representation spaces in panel **a**, found by a variety of methods. Runtimes to calculate the full matrix are shown above each method, except for Jensen-Shannon divergence (JSD) because it required training an additional classification network on top of each latent space. The stochastic shape metric requires the dimensionality of the compared spaces to match; undefined entries are grayed out. The Spearman rank correlation between similarity measures is shown in the bottom right.

We additionally observe differences between the methods in relative similarities between the latent spaces. All methods except linear CKA and stochastic shape metrics detect the similarity of the quasi-one-dimensional encodings (representation spaces **i**, **iv**, and **vi**). These encodings are not related by simple transformations, though a downstream neural network can extract similar information from each (see JSD). Representation spaces **vii** and **ix** are similar by information content, as variations of splitting the data into two groups, but they are deemed different by all but NMI, VI, and the Bhattacharyya-based CKA. While stochastic shape metrics satisfy the desirable properties of a metric, the generalized NMI/VI offer a measure of similarity more relevant for downstream processing owing to their utilization of mutual information.

## 4.2 Unsupervised detection of structure: channel similarity

We next analyzed the consistency of information fragmentation in ensembles of generative models trained on image datasets. Using fifty models with ten latent dimensions (channels) each, for a variety of datasets, methods, and hyperparameters (some released with Locatello et al. (2019)[1]), we assessed structure in an ensemble's channels. Every latent dimension of every model in an ensemble was compared pairwise to every other; we used the Bhattacharyya fingerprint approach with a sample of 1000 images randomly selected from the training set. Before using OPTICS to group latent dimensions by similarity (described in Sec. 3.3), we removed dimensions transmitting less than 0.01 bits of information. We found NMI to more successfully detect channels with similar information content, and use it in this section (comparison with VI in Appx. E).

In Fig. 3, the matrices display the pairwise NMI between all informative dimensions in the ensemble, and have been reorganized by OPTICS such that highly similar latent dimensions are close together and appear as blocks along the diagonal. On the left of each similarity matrix is the NMI with the ground truth generative factors (or other label information), and above is the OPTICS reachability profile where identified groupings of consistently learned channels are indicated with shading and Roman numerals.

In Fig. 3a, the regularization of a $\beta$-VAE is increased for the `dsprites` dataset (Higgins et al., 2017). Although there are more channels that convey information for lower regularization ($\beta=4$), there is little discernible structure in the population of channels aside from a group of channels that communicate roughly the same information about `scale`. With $\beta = 16$, a block diagonal structure is found, and channels with the same `xpos`, `ypos`, and `scale` information are found repeatedly across runs in the ensemble. We applied the same analysis to an ensemble of InfoGAN-CR models (Lin et al., 2020), whose models additionally encoded `shape` information consistently. We approximated the latent distribution for an image by first encoding it, then re-generating 256 new images with the predicted latent dimensions fixed and the remaining (unconstrained) dimensions randomly resampled, and finally using the moments of the newly predicted latent representations as parameters for a Gaussian distribution. We note that in any scenario where a natural probability distribution exists per datum in the representation space, the method can be used.

Fig. 3b shows remarkable consistency of information fragmentation by an ensemble of $\beta$-VAEs trained on the `cars3d` dataset (Reed et al., 2015), though **not** with regards to the provided generative factors. The `object` factor is broken repeatedly into the same set of information fragments, which is sensible given that it is a fine-grained factor of variation that comprises many lower-entropy factors such as color and height. More surprising is that three of the fragments contain a mix of camera pose information and `object` information that is consistently learned across the ensemble. The majority of existing disentanglement metrics rely on ground truth generative factors—including the FactorVAE score (Kim & Mnih, 2018), mutual information gap (Chen et al., 2018), DCI (Eastwood & Williams, 2018), and InfoMEC (Hsu et al., 2023)—and would miss the consistency found when comparing the information content of individual latent dimensions.

Given a group of latent dimensions identified by OPTICS, we take as a representative the dimension that maximizes the average similarity to others in the group. Latent traversals visualize the fragmented information: groups **i** and **iii** communicate different partial information about both pose and color, while group **vi** conveys information about the color of the car with no pose information.

Is this particular way of fragmenting information about the `cars3d` dataset reproduced across different methods? In Fig. 3c we compared the information content of the seven representative latent dimensions from the $\beta$-VAE ($\beta = 16$) to the OPTICS-reorganized latent dimensions for different methods ($\beta$-TCVAE (Chen

---

[1]https://github.com/google-research/disentanglement_lib/tree/master

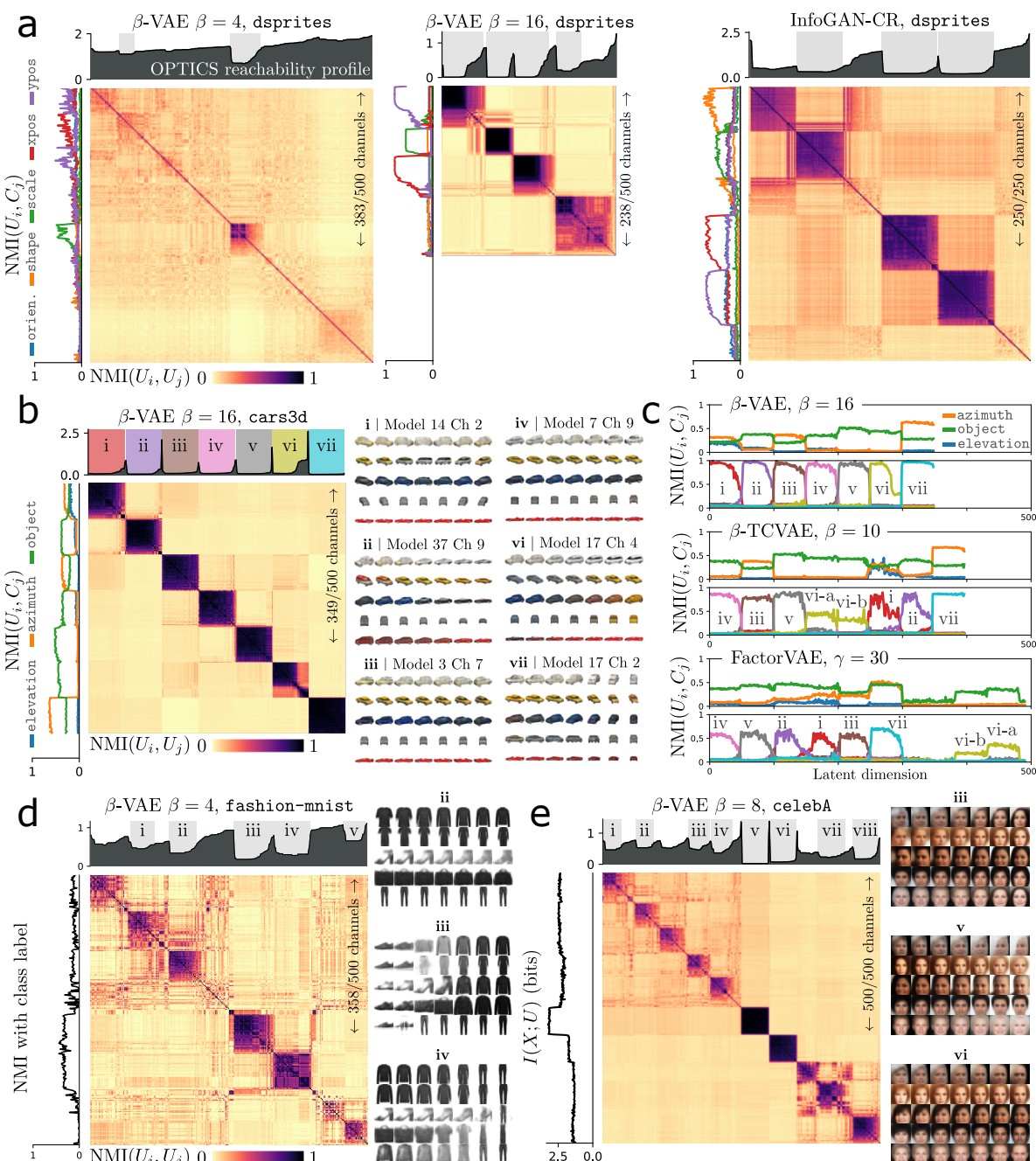

Figure 3: **Assessing the consistency of channel information in ensembles of models.** We used NMI as a similarity measure for OPTICS to detect fragments of information that are consistently stored in individual channels in an ensemble of trained models. **(a)** The channel consistency of models trained on `dsprites`, for $\beta$-VAE and InfoGAN-CR. The information with respect to generative factors is shown on the left of each similarity matrix. The $\beta = 4$ $\beta$-VAE fragmented information inconsistently compared to the other two ensembles. **(b)** For a $\beta$-VAE ensemble trained on `cars3d`, the information content of channels was highly consistent, with seven distinct combinations of the three generative factors. Latent traversals for a representative channel from each grouping visualize the information content. **(c)** We compare the information content of the representatives from panel **b** to that of channels in $\beta$-TCVAE and FactorVAE ensembles. **(d,e)** Channel similarity and latent traversals for $\beta$-VAE ensembles trained on `fashion-mnist` and `celebA`. Additional channel similarity analyses and latent traversals can be found in Appx. A.

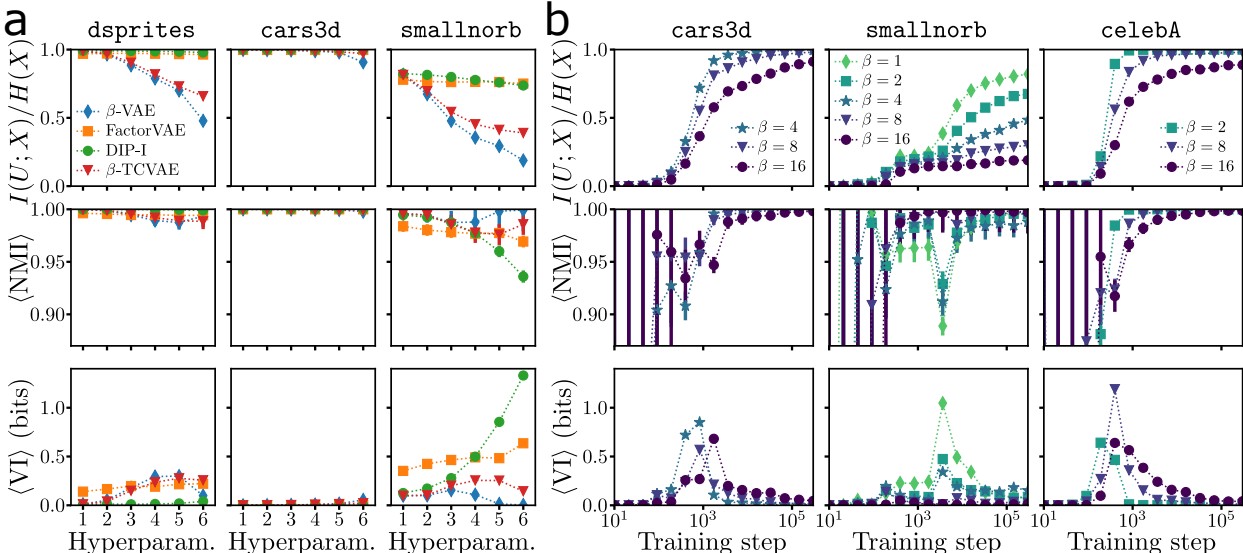

Figure 4: **Comparing full latent spaces. (a)** We compare trained models across several methods and six hyperparameters each, all from Locatello et al. (2019). **(b)** We compare $\beta$-VAE models over the course of training. $I(U;X)/H(X)$ is the fraction of total information about the dataset contained in the latent space; a value of one means all data points are well-separated in the latent space. $\langle NMI \rangle$ and $\langle VI \rangle$ denote the average pairwise NMI and VI values over five models in an ensemble. All mutual information terms were estimated via Monte Carlo, and the displayed error bars are the standard error after accounting for the uncertainty on the constituent mutual information terms.

et al., 2018), $\beta = 10$ and FactorVAE (Kim & Mnih, 2018), $\gamma = 30$). The consistent fragments for the other methods are recognizable from those of the $\beta$-VAE ensemble, though interestingly the FactorVAE conveyed comparatively less `azimuth` information in several of the fragments. Channels of group **vi** for the $\beta$-VAE jointly encoded information about the tint of the windows and the color of the car, and this information was encoded separately by the $\beta$-TCVAE and FactorVAE (traversals in Appx. A).

Finally, we studied the manner of information fragmentation on datasets which are not simply an exhaustive set of combinations of generative factors (Fig. 3d,e). For $\beta$-VAE ensembles trained on `fashion-mnist` (Xiao et al., 2017) and `celebA` (Liu et al., 2015), some information fragments are more consistently learned than others. A particular piece of information that distinguishes between shoes and clothing (group **iii**) was repeatedly learned for `fashion-mnist`; for `celebA`, a remarkably consistent fragment of information conveyed background color (group **v**). The remaining information was less consistently fragmented across the ensemble.

### 4.3 Assessing the content of the full latent space

If an ensemble of models fragments information into channels inconsistently (e.g., Fig. 3a), is it because the information content of the full latent space varies across runs? We compared the information contained in the full 10-dimensional latent spaces of five models from each VAE ensemble (Fig. 4). The amount of information was generally too large for estimation via Bhattacharyya matrices, which saturates at the logarithm of the fingerprint size, so we used Monte Carlo estimates of the mutual information. Error bars in Fig. 4 are the propagated uncertainty from the mutual information measurements, with additional details in Appx. C. Whereas NMI proved more useful for the channel similarity analysis, VI tended to be more revealing about the heterogeneity of full latent spaces. Due to the form of its normalization, NMI becomes unreliable—i.e., is plagued by large uncertainty—when the information contained in the latent space $I(U;X)$ is small.

Across methods and hyperparameters for each method, the information contained in the full latent space was largely consistent over an ensemble (Fig. 4a; models from Locatello et al. (2019)). For the `cars3d` dataset, most latent spaces conveyed the full entropy of the dataset, meaning all representations were well-separated

for the dataset of almost 18,000 images, making the information content trivially similar. By contrast, none of the models trained on `smallnorb` contained more than around 80% of the information about the dataset, and the similarity of latent spaces across members of an ensemble was more dependent on method and hyperparameter. Ensembles of $\beta$-VAE and $\beta$-TCVAE were fairly consistent even though the transmitted information dropped considerably with increasing $\beta$; DIP-I (Kumar et al., 2018) showed the opposite behavior, where consistency varied strongly with $\lambda_{od}$ but transmitted information did not.

How does the consistency of latent space information evolve over the course of training? For $\beta$-VAE ensembles, we again compared the full latent spaces for five models with random initializations (Fig. 4b). For all datasets considered (`cars3d`, `smallnorb`, `celebA`), consistency across the models dropped at around the same point in training that total information increased dramatically; after this point, all models encapsulated nearly the same information in their latent spaces up to convergence.

### 4.4 Model fusion in a toy example

Finally, we demonstrate the capacity for model fusion with the proposed measures of representation space similarity. Consider a dataset with a single generative factor with SO(2) symmetry, such as an object's hue. Difficulties can arise when the global structure of a degree of freedom is incompatible with the latent space (Falorsi et al., 2018; Zhou et al., 2019; Esmaeili et al., 2024). In this example, a one-dimensional latent space cannot represent the global circular topology of SO(2), leading to discontinuities in the representation.

To clearly demonstrate the synthesis of information from multiple representation spaces, we trained an ensemble of $\beta$-VAEs, each with a one-dimensional latent space that was insufficient to represent the global structure of the generative factor. Fig. 5a shows an example latent space and its associated distinguishability matrix, showing pairwise Bhattacharyya coefficients between posterior distributions. The matrix reveals flaws in the latent space where similar values of the generative factor have dissimilar representations.

Assuming the flaws are randomly distributed from one training run to the next—which need not be true—the fusion of multiple such latent spaces might yield an improved representation of the generative factor. We performed gradient descent directly on posterior distributions in a two-dimensional latent space, with the objective to maximize average similarity with the ensemble. Namely, we computed either NMI or the $\exp(-\text{VI})$ using the Bhattacharyya matrix corresponding to the optimizable representations—recalculated every training step—and those of the ensemble. As shown in Fig. 5b,c, the synthesized latent space more closely captured the global structure of the generative factor as the ensemble size grew. To quantify the performance, we employed the continuity metric used in Esmaeili et al. (2024) and adapted from Falorsi et al. (2018), substituting the Bhattacharyya distance between posteriors for the Euclidean distance typically used for point-based representations (Appx. F).

Both NMI and VI proved effective in boosting an ensemble of weak representation spaces to represent the generative factor, with fidelity (measured by continuity) improving for larger ensembles. By comparison, directly maximizing the mutual information between the synthesis space and the ensemble, $\langle I(U; V_i) \rangle_i$, resulted in scattered representations with no overlap (Fig. 5c, right). The normalization terms of NMI and VI were essential for preserving the relational structure between data points. Finally, it is worth noting that after training the ensemble of weak models, neither the original data nor the models were needed to train the synthesis space: the latent representations were optimized directly from the Bhattacharyya matrices.

## 5 Discussion

The processing of information can be directly assessed when representation spaces are probabilistic because information theoretic quantities are well-defined. In this work, we generalized two classic measures for comparing the information content of clustering assignments to make them applicable to probabilistic spaces. By focusing on the information content of a representation space, we can assess what information is available for downstream processing, while remaining agnostic to aspects of the spaces such as their dimensionality, their discrete or continuous nature, and even whether a space serves as auxiliary information like labels or annotations (Newman & Clauset, 2016; Savić, 2018; Bazinet et al., 2023). While the examples in this work focused on relatively low-dimensional latent spaces that are common in practice, scaling to higher-dimensional

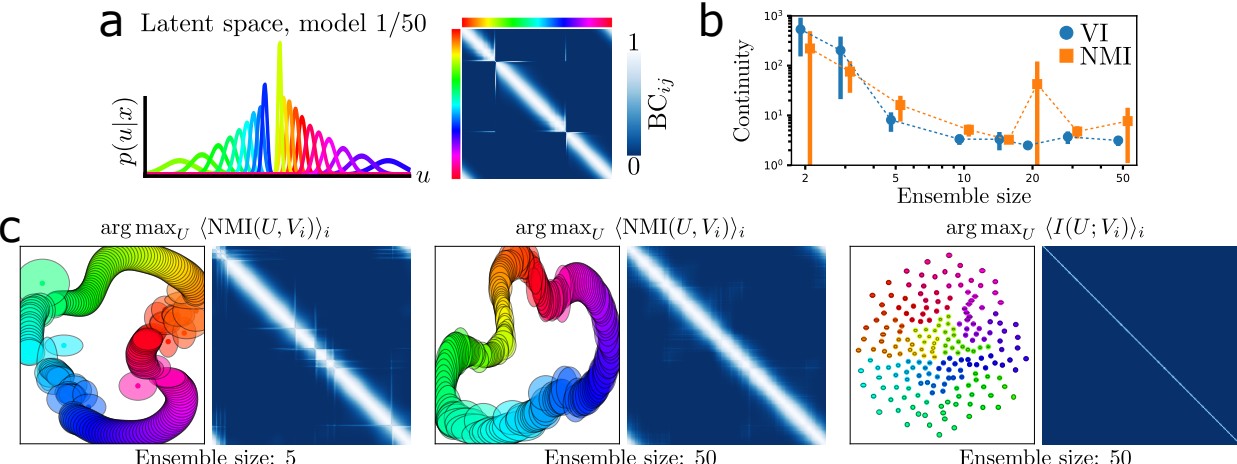

Figure 5: **Fusing weak representation spaces. (a)** Example of a one-dimensional latent space of a $\beta$-VAE trained on a dataset generated from a single periodic factor (color hue), which has SO(2) symmetry. The latent space exhibits flaws where similar values of the generative factor are mapped to dissimilar representations, as seen in the posterior distributions (*left*) and the distinguishability matrix of Bhattacharyya coefficients between posteriors, $BC_{ij}$ (*right*). **(b)** We optimized a synthesis representation space to maximize similarity with an ensemble of such one-dimensional latent spaces. The continuity of statistical distances between neighboring points, an assessment of the fidelity of the global structure of the generative factor, improved as the ensemble size grew. Error bars show the standard deviation over five experiments, and values are offset horizontally for visibility. **(c)** Synthesized two-dimensional representation spaces (posterior means shown as points; covariances as shaded ellipses) and their corresponding distinguishability matrices. Panels compare results when maximizing average NMI (*left*, *middle*) and mutual information (*right*).

representation spaces may face challenges related to the reliable estimation of mutual information (McAllester & Stratos, 2020).

With differentiable formulations for NMI and VI, model fusion and the more general problem of representational alignment (Sucholutsky et al., 2023; Muttenthaler et al., 2024) can be effectively approached from an information theoretic perspective. Potential applications include aligning representation spaces across models trained on different subsets of data, improving ensemble methods, and evaluating consistency of representations in multitask learning or domain adaptation, where reconciling heterogeneous latent spaces is often crucial.

A more subtle contribution of this work is to shift the current focus of unsupervised disentanglement evaluation. As is clear from the `cars3d` information fragmentation (Fig. 3b), existing metrics that compare latent dimensions to ground truth generative factors can completely miss consistent information fragmentation. Unsupervised methods of evaluation (Duan et al., 2020; Lin et al., 2020) assess consistency of fragmentation at the scale of models, which can obscure much about the manner of fragmentation. Consider the fragmentation of information about the `celebA` dataset in Fig. 3e: an assessment of the similarity of models would find middling values across the ensemble and miss that two fragments of information are remarkably consistent. We argue that more fine-grained inspection of information fragmentation is essential for a deeper understanding of disentanglement in practice.

The current work largely focused on comparing repeat training runs in an ensemble, making computational costs a consideration. We found 50 models per ensemble to be sufficient to assess channel similarity, and 5 models per ensemble for the full latent space. We also found that there is much to learn from ensembles of relatively simple models. On our machine, training a single model from a recently proposed method (QLAE, Hsu et al. (2023)) took more than five hours, and in the same amount of time we could train ten $\beta$-VAEs with a simpler architecture. Model fusion with weak models that are inexpensive to train, as in Sec. 4.4, might offer a promising alternative to representation learning with more computationally expensive models.

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

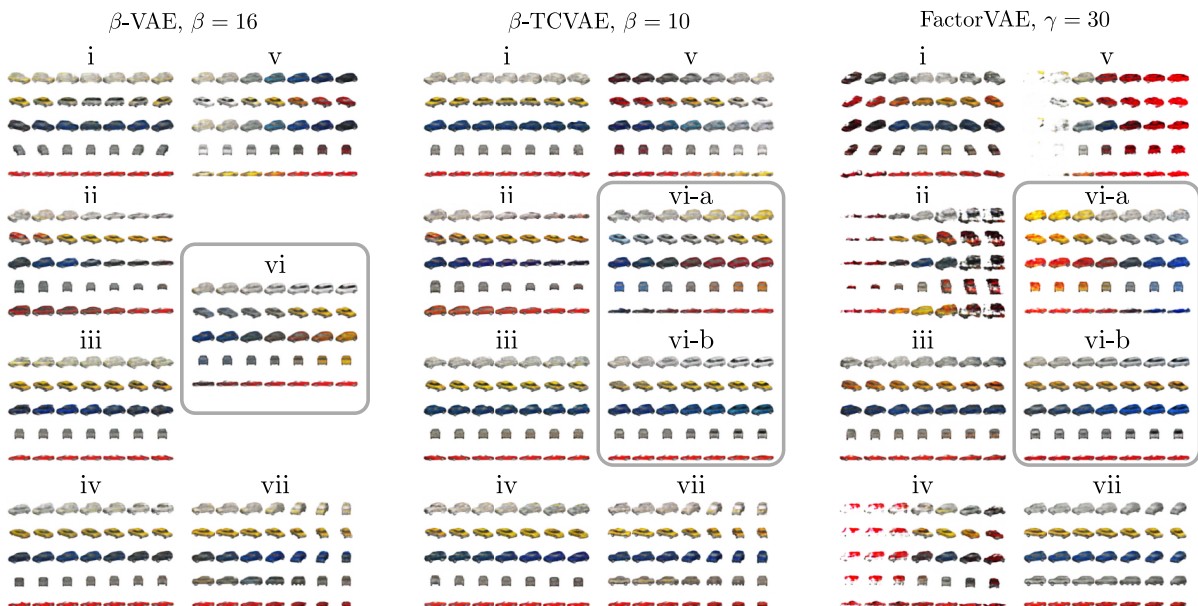

Figure 6: **Latent traversals for `cars3d` groups from Fig. 3b,c.** We traverse the representative channels for the groups found by OPTICS, and reorder them to align with the ordering for $\beta$-VAE (left). Note that group **vi** splits in two groups for both the $\beta$-TCVAE and the FactorVAE. The tint of the windows and the color of the car were encoded jointly for the $\beta$-VAE, and then separately for the other two methods. Traversals are over the range $[-2, 2]$.

## A    Appendix: Extended channel similarity results

In Fig. 6 we present additional latent traversals for the `cars3d` channel groupings presented in Fig. 3c. With the channels most centrally located in each group (as before), we also display latent traversals for $\beta$-TCVAE ($\beta$=10) and FactorVAE ($\gamma = 30$).

Fig. 7 visualizes the channel similarity structure on `mnist` (LeCun et al., 2010) and `fashion-mnist`, with latent traversals for four channels per grouping to show the consistency of the encoded information.

In Figs. 8, 9, and 10, we repeat the structural analysis of Sec. 4.2 with all $\beta$-VAE hyperparameters explored by the authors of Locatello et al. (2019). The effect of increasing $\beta$ is clearly observed by the emergence of block diagonal channel similarity matrices, though with fewer informative channels for increased $\beta$.

Consider the `cars3d` ensembles in Fig. 8. With $\beta = 1, 2$, the information fragmentation is not consistent across repeat runs, even though the amount of information shared with the three generative factors is fairly consistent. This highlights the value of directly comparing the information content of channels with NMI or VI instead of comparing indirectly via information content about known generative factors, with the added benefit of being fully unsupervised. For $\beta = 4$ the learned fragments of information start to coalesce, with information regularization breaking the degeneracy that plagues unsupervised disentanglement (Locatello et al., 2019). For $\beta = 8, 16$, the regularization is strong enough to form consistent fragments of information across random initializations.

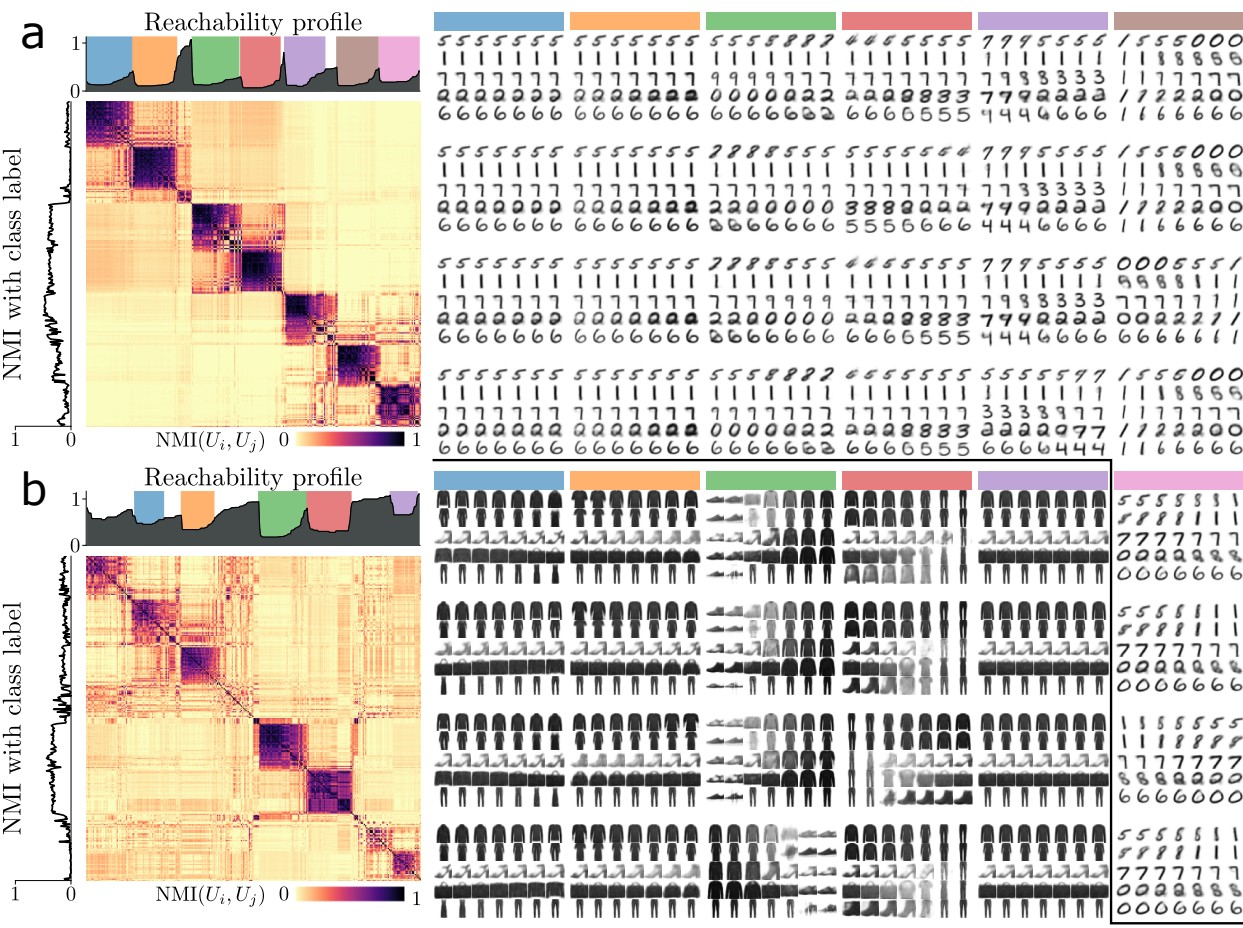

Figure 7: **Channel similarity structure on MNIST and Fashion-MNIST.** Channel similarity analysis for 50 $\beta$-VAEs trained on **(a)** MNIST ($\beta=8$), and **(b)** Fashion-MNIST ($\beta = 4$). The most central four channels to each of the found groupings (indicated by colors) are visualized via latent traversal on the right.

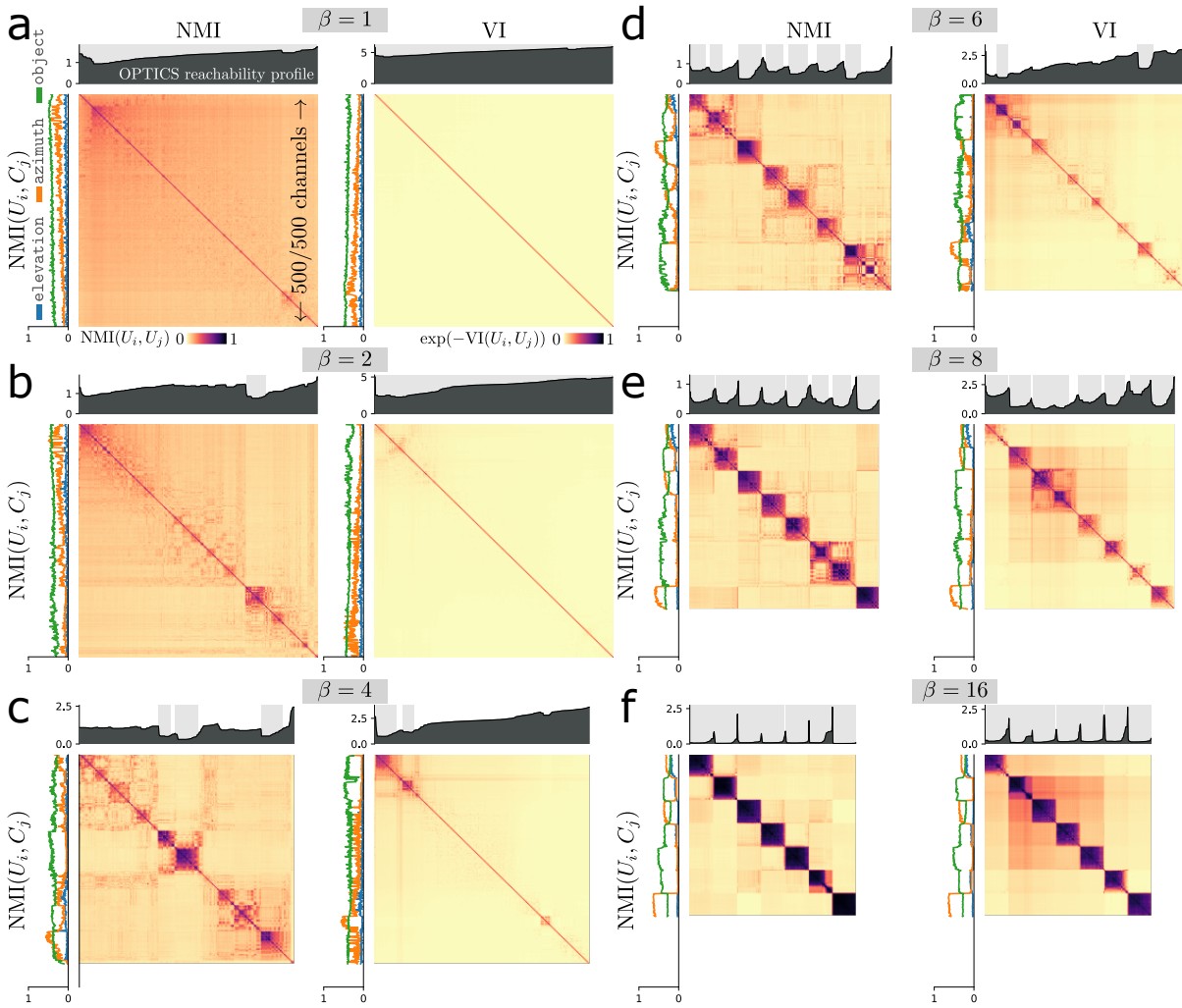

Figure 8: **Channel similarity structure for cars3d, $\beta$-VAE, assessed with NMI and VI. (a-f)** Ensembles with values of $\beta$ from 1 to 16, with channels reordered using OPTICS and either NMI (left) or VI (right) for the method of comparison. Regardless of the measure used for comparison, the NMI with the generative factors is shown on the left of each pairwise similarity matrix. All matrices are sized to $500 \times 500$, with uninformative channels ($I(U_i; X) \leq 0.01$ bits) displayed as white.

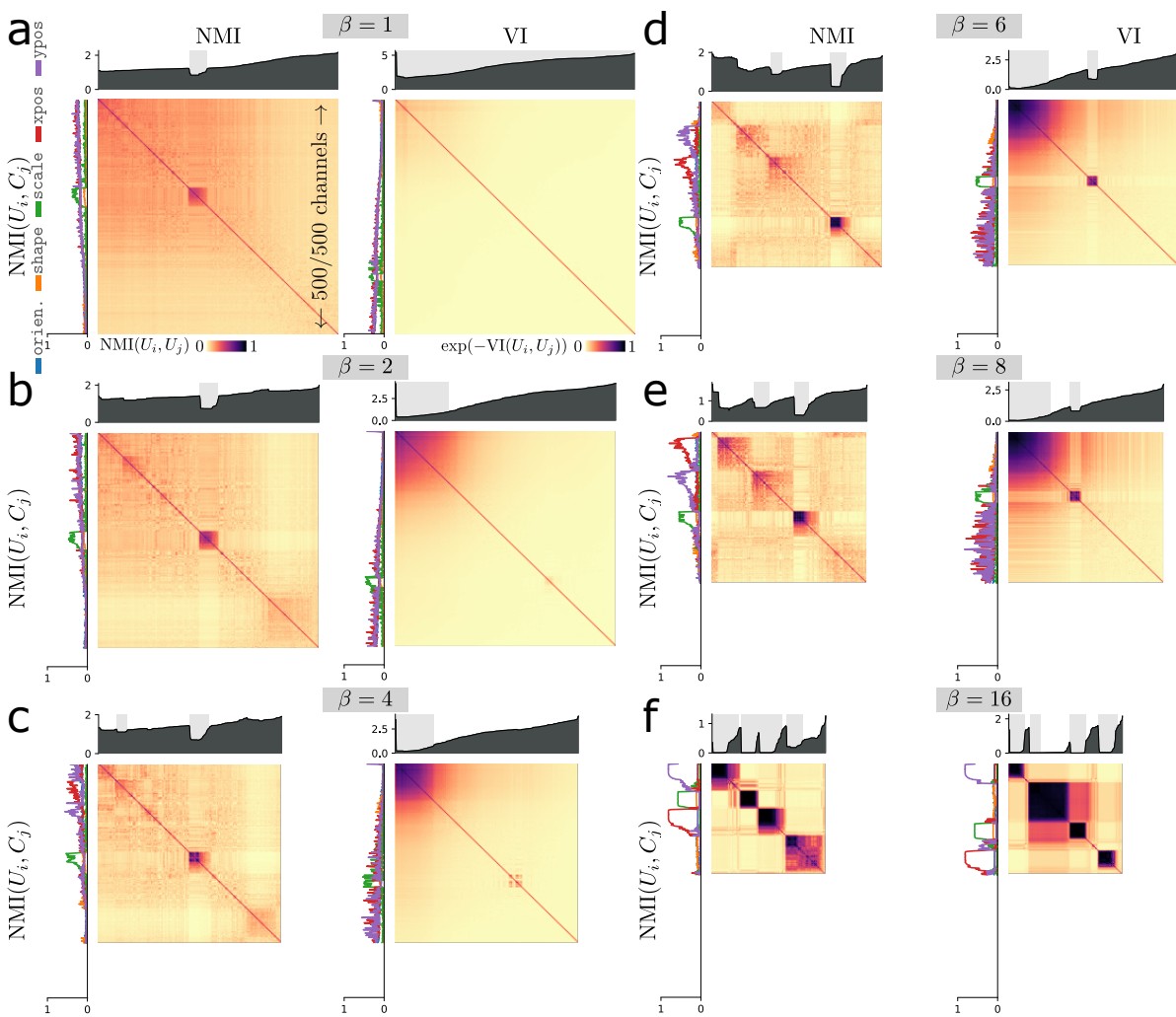

Figure 9: **Channel similarity structure for** `dsprites`, $\beta$-**VAE, assessed with NMI and VI.** Everything in this figure mirrors Fig. 8.

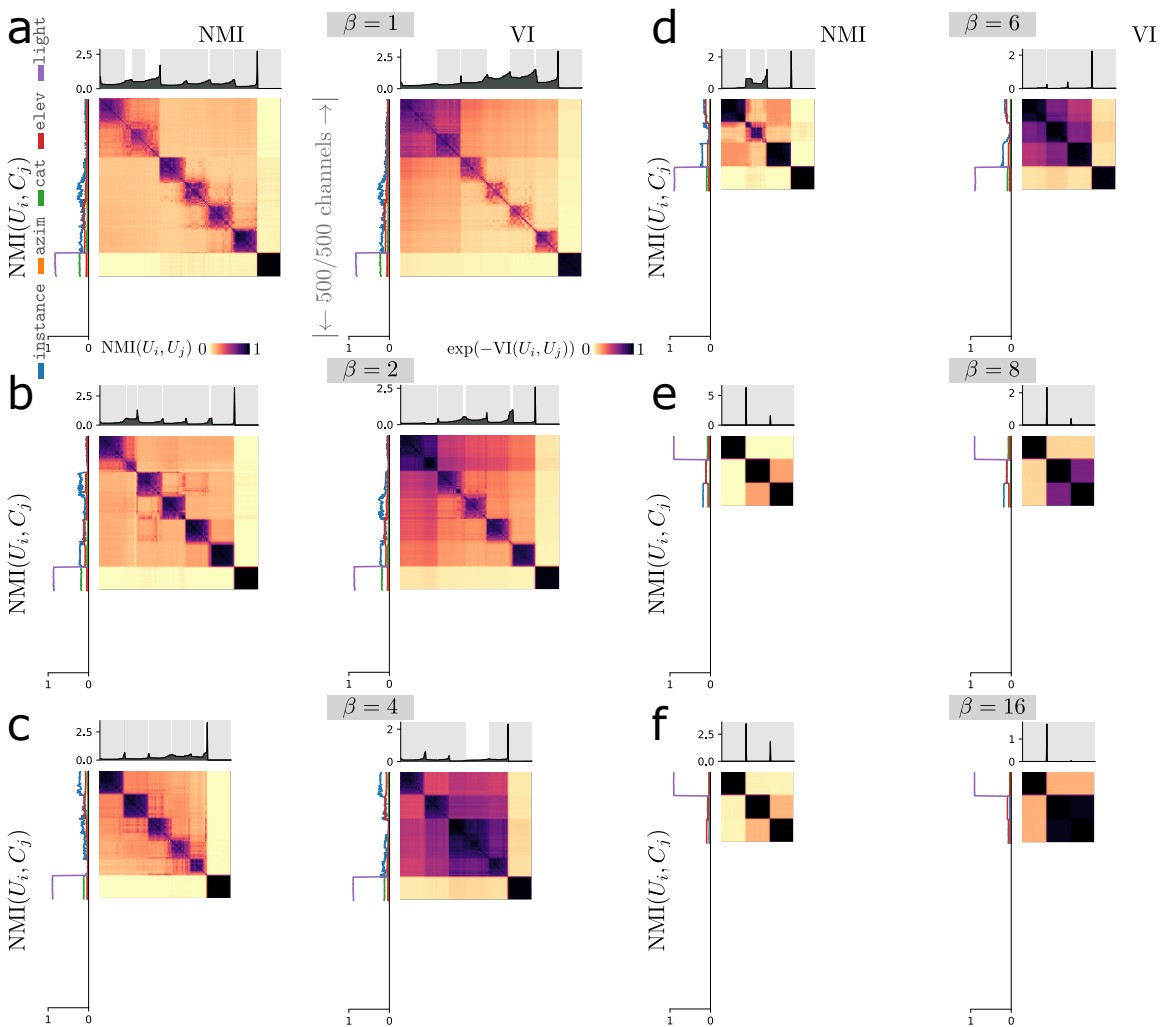

Figure 10: **Channel similarity structure for `smallnorb`, $\beta$-VAE, assessed with NMI and VI.** Everything in this figure mirrors Fig. 8.

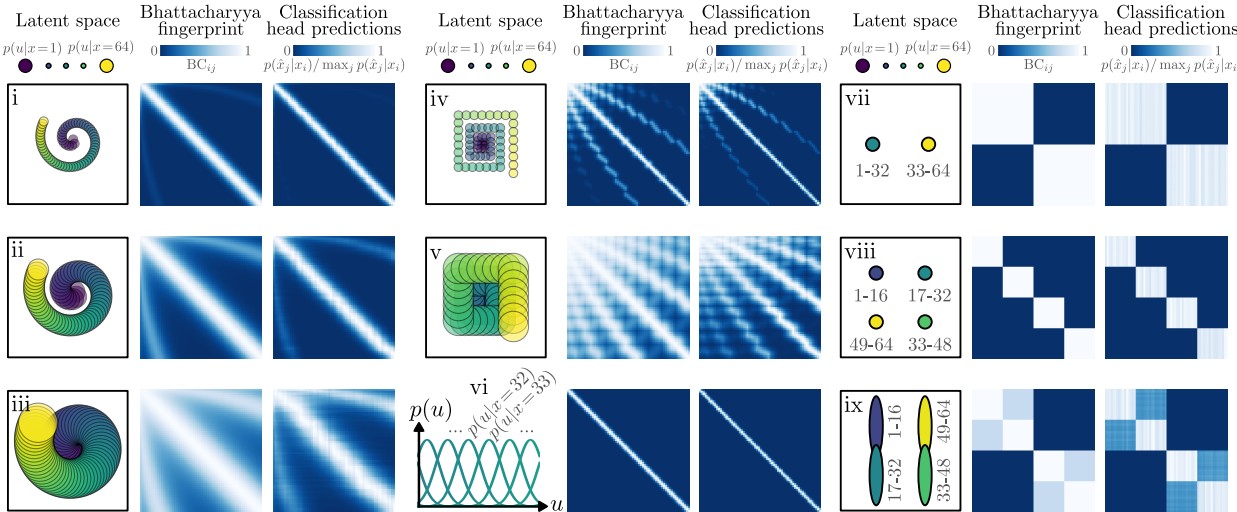

Figure 11: **Full comparison of Bhattacharyya fingerprints and classification outputs from latent spaces of Fig. 2.**

## B   Appendix: Extended results on synthesized latent space comparison (Sec. 4.1)

We include in Fig. 11 the full set of comparisons between the Bhattacharyya fingerprint and the average predictions of a trained classification head for the nine synthetic latent spaces of Sec. 4.1. Fig. 12 shows the pairwise scatter plots for the methods of comparing the latent spaces.

## C   Appendix: Information estimation using Bhattacharyya distinguishability matrices and Monte Carlo estimator

To estimate the mutual information $I(U; X)$ from the Bhattacharyya distinguishability matrices, we have employed the lower bound derived in Kolchinsky & Tracey (2017) for the information communicated through a channel about a mixture distribution (following the most updated version on arXiv[2]). The bound simplifies greatly when the empirical distribution is assumed to be a reasonable approximation for the data distribution, and then we further assume the sample of data used for the fingerprint allows for an adequate approximation of the marginal distribution in latent space. First we reproduce the bound from Sec. V of Kolchinsky & Tracey (2017) using the notation of this work, and then we describe our assumptions to apply the bound as an estimate of the information contained in a probabilistic representation space.

Let $X$ be the input to a channel, following a mixture distribution with $N$ components, $x \sim p(x) = \sum_{i=1}^{N} c_i p_i(x)$, and $U$ the output of the same channel, $u \sim p(u) = \sum_{i=1}^{N} c_i \left( \int_{\mathcal{X}} p(u|x) p_i(x) dx \right)$. Then we have

$$I(X; U) \geq -\sum_i c_i \ln \sum_j c_j BC_{ij} + H(U|C) - H(U|X), \tag{6}$$

where $C$ is a random variable representing the component identity.

In this work, we assume that the data distribution can be approximated by the empirical distribution, $p(x) \approx \sum_i^N \delta(x - x_i)/N$, simplifying Eqn. 6 so that $c_i \equiv 1/N$ and $H(U|C) = H(U|X)$ because the identity of the component is equivalent to the identity of the datum. Finally, we assume that the set of posterior distributions for a representative sample of size $M$ of the dataset, taken for the fingerprint, adequately approximates the empirical distribution. Larger samples may be necessary in different scenarios when the

---

[2]https://arxiv.org/abs/1706.02419

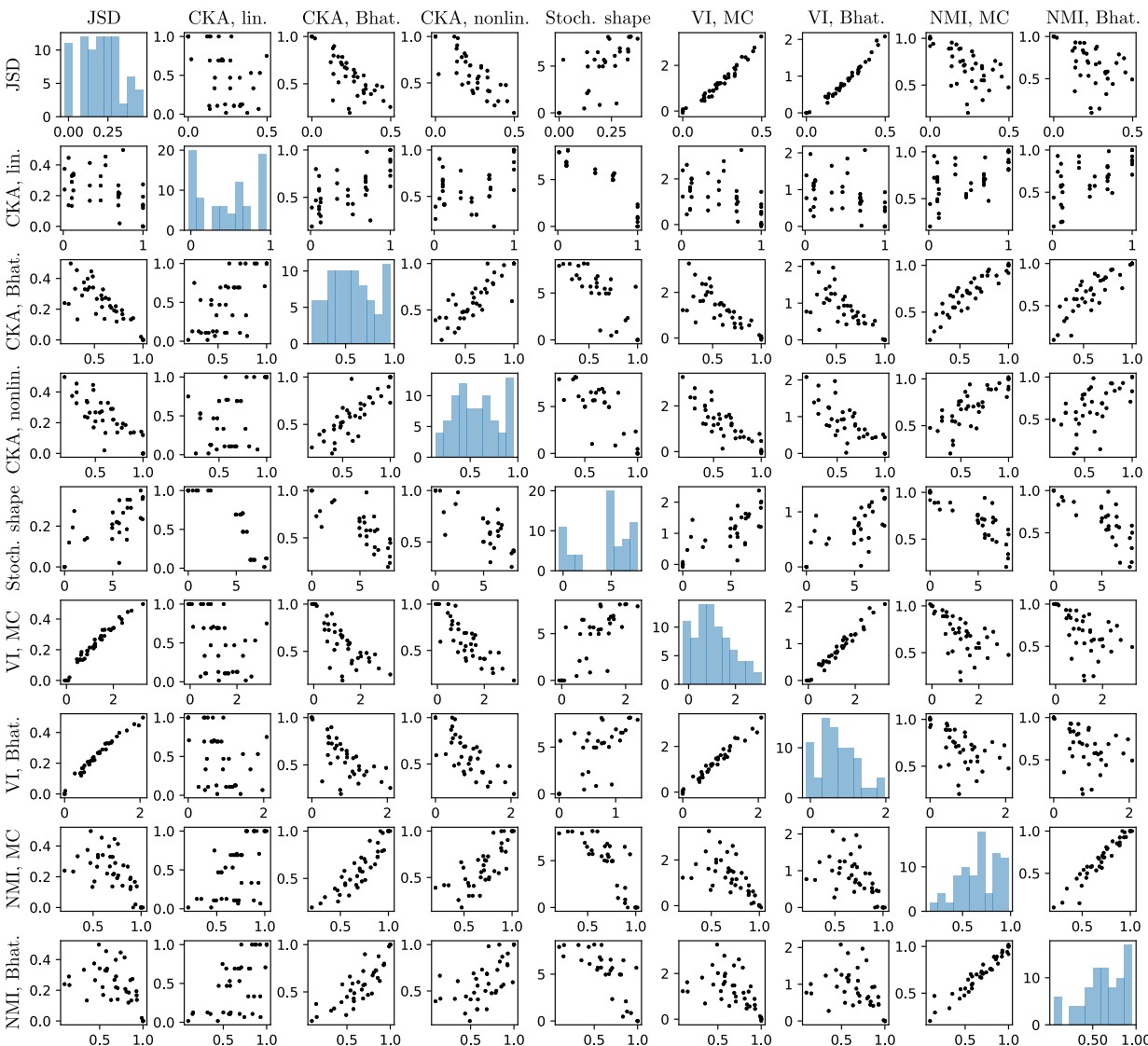

Figure 12: **Pairwise scatter plot comparison of similarity measures from Fig. 2.**

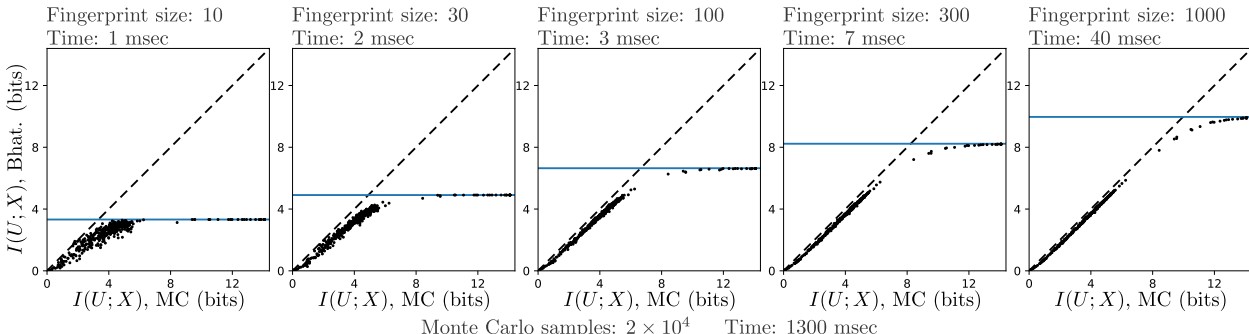

Figure 13: **Information estimates with Bhattacharyya fingerprints and Monte Carlo.** For 500 channels randomly sampled from across all models released by Locatello et al. (2019) for the `cars3d` dataset (including all methods and all hyperparameters), we estimated the amount of information transmitted by each channel $I(U;X)$ using the Bhattacharyya matrix fingerprints and Monte Carlo (MC) sampling. Error bars are displayed for the MC estimates, though they are generally smaller than the markers. The dashed black line represents equality between the two estimates, and the solid blue line is the logarithm of the fingerprint size, which is the saturation point for the Bhattacharyya estimate. Listed run times are for a single channel, excluding the time to load models but including inference and the calculation necessary for $I(U;X)$.

amount of information transmitted by channels is larger than a handful of bits, but $M = 1000$ appeared sufficient for the analyses of this work.

The generalizations of NMI and VI required the information conveyed by two measurements from different channels, $I(X;U,V)$ as well as from the same channel, $I(X;U,U')$. The matrix of Bhattacharyya coefficients given measurements $U$ and $V$ is simply the elementwise product of the coefficients given $U$ and the coefficients given $V$. The posterior in the joint space of $U$ and $V$ is factorizable given $x$—i.e., $p(u,v|x) = p(u|x)p(v|x)$—because the stochasticity in each channel is independent. The same is true for the joint space of $U$ and $U'$, two draws from the same channel. The Bhattacharyya coefficient of the joint variable simplifies,

$$
\begin{aligned}
\mathrm{BC}_{ij}^{UV} &= \int_{\mathcal{U}} \int_{\mathcal{V}} \sqrt{p(u,v|x_i)p(u,v|x_j)} du dv \\
&= \int_{\mathcal{U}} \sqrt{p(u|x_i)p(u|x_j)} du \int_{\mathcal{V}} \sqrt{p(v|x_i)p(v|x_j)} dv \\
&= \mathrm{BC}_{ij}^{U} \times \mathrm{BC}_{ij}^{V}.
\end{aligned}
\tag{7}
$$

For the Monte Carlo estimator, we sampled random data points from the dataset $x \sim p(x)$ and then a random embedding vector from each posterior distribution, $u \sim p(u|x)$. Then we computed the log ratio of the likelihood under the corresponding posterior $p(u|x)$ and the aggregated posterior $p(u)$, which was gotten by iterating over the entire dataset. We repeated $N$ times to estimate the following expectation:

$$
I(X;U) = \mathbb{E}_{x \sim p(x)} \mathbb{E}_{u \sim p(u|x)} \left[ \log \frac{p(u|x)}{p(u)} \right]
\tag{8}
$$

For the analysis of Fig. 4, $N = 2 \times 10^5$ for all datasets except `dsprites` and `celebA`, where $N = 6 \times 10^4$; the standard error of the estimate was on the order of 0.01 bits or less.

Computing the fully aggregated posterior is time consuming; could a subsample of the dataset be used instead? In Fig. 14, we varied the fraction of the dataset used to compute the (partially) aggregated posterior for a full 10-dimensional latent space for three models that transmit an intermediate amount of information about the dataset on which they were trained. Each point represents the mutual information estimate following Eqn. 8 for a random subset of the dataset, and the extent of its error bar is given by the standard error: the standard deviation of the set of values divided by the square root of the size of the set. We find that the transmitted information is relatively robust to using half of the dataset when computing the aggregated posterior, but

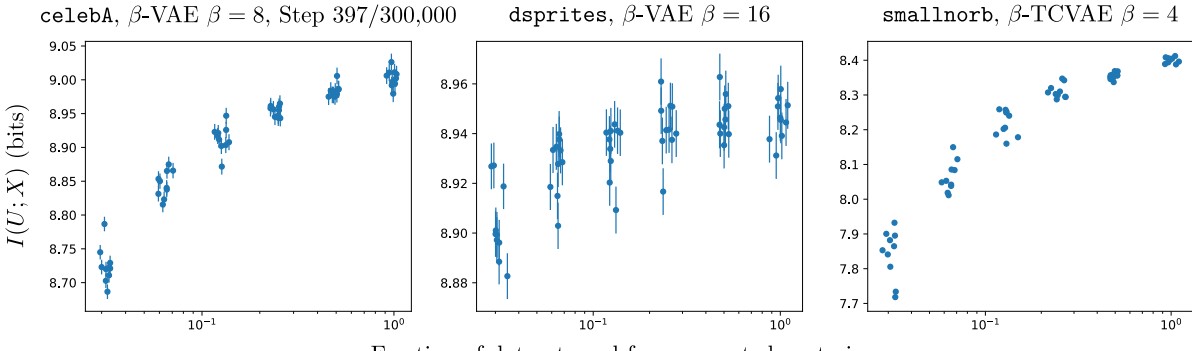

Figure 14: **Partially aggregated posterior for the Monte Carlo estimator.** For three models selected from those of Fig. 4, we seleced random subsets of the full dataset to use when computing the aggregated posterior for estimating the transmitted information $I(U;X)$ of the full 10-dimensional latent space. The points are offset horizontally by random amounts for visibility, and the error bars show the standard error on the mean of the Monte Carlo samples.

deleterious effects grow when reducing the dataset by a factor of 10 or more. Interestingly, the `dsprites` estimate was largely insensitive to using 3% of the dataset, perhaps due to its highly structured nature.

In Fig. 4, the NMI and VI values are averaged over all pairwise comparisons for the five representation spaces, with weights given by the inverse squared uncertainty on each pairwise value. The uncertainty was calculated by propagating the uncertainty of Monte Carlo estimates of $I(X;U)$ according to the expression for VI or NMI. To be specific, as $\mathrm{VI}(U,V) = 2I(X;U,V) - I(X;U,U') - I(X;V,V')$, the propagated uncertainty is given by

$$\Delta\mathrm{VI}(U,V) = \sqrt{4\Delta I(X;U,V)^2 - \Delta I(X;U,U')^2 - \Delta I(X;V,V')^2}, \tag{9}$$

with $\Delta I(X;U,V)$ the standard error from the Monte Carlo estimate. The expression for the uncertainty on the NMI estimate contains many more terms, and we will not reproduce them here. Finally, the error bars in Fig. 4 are the standard error of the weighted mean,

$$\Delta Q = \left(\sum_i^n (\Delta Q_i)^{-2}\right)^{-\frac{1}{2}}, \tag{10}$$

for quantity $Q$.

## D  Appendix: Broken triangle inequality for the generalized VI

Here we provide an example of three clusterings that breaks the triangle inequality for the generalized VI. Thus, while VI for hard clusters satisfies the properties of a metric (Crutchfield, 1990), the generalization to soft clusters does not.

In Fig. 15, there are three clusterings ($U$, $V$, and $W$) of four equally likely data points ($x \in \{a, b, c, d\}$). $U$, $V$, and $W$ each communicate one bit of information about $X$, but $U$ and $W$ are hard clusterings while $V$ is a soft clustering. We have $I(U;V) = I(V;W) = 0.5$ bits, $I(U;W) = 0$ bits, $I(V;V') = 0.5$ bits, and $I(U;U') = I(W;W') = H(U) = 1$ bit. The generalized VI from $U$ to $V$ and then from $V$ to $W$ is less than from $U$ to $W$ directly.

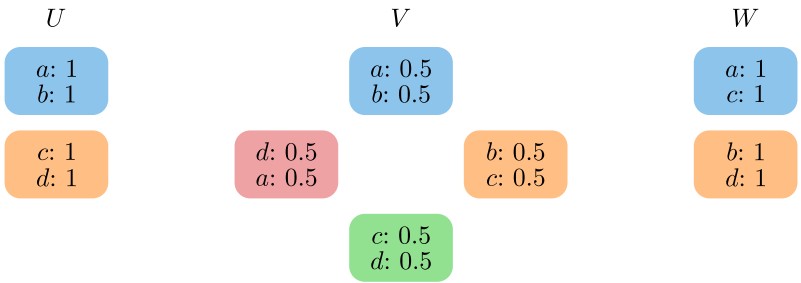

Figure 15: **Example that breaks the triangle inequality for generalized VI.** Four datapoints $a, b, c, d$ are clustered according to scheme $U$, $V$, or $W$. The total VI from $U$ to $V$ and then $V$ to $W$ is less than the VI from $U$ to $W$.

$$\mathrm{VI}(U, W) = I(U; U') + I(W; W') - 2I(U; W)$$
$$= 2 \text{ bits}$$
$$\mathrm{VI}(U, V) + \mathrm{VI}(V, W) = 2\left(I(U; U') + I(V; V') - 2I(U; V)\right)$$
$$= 1 \text{ bit.}$$

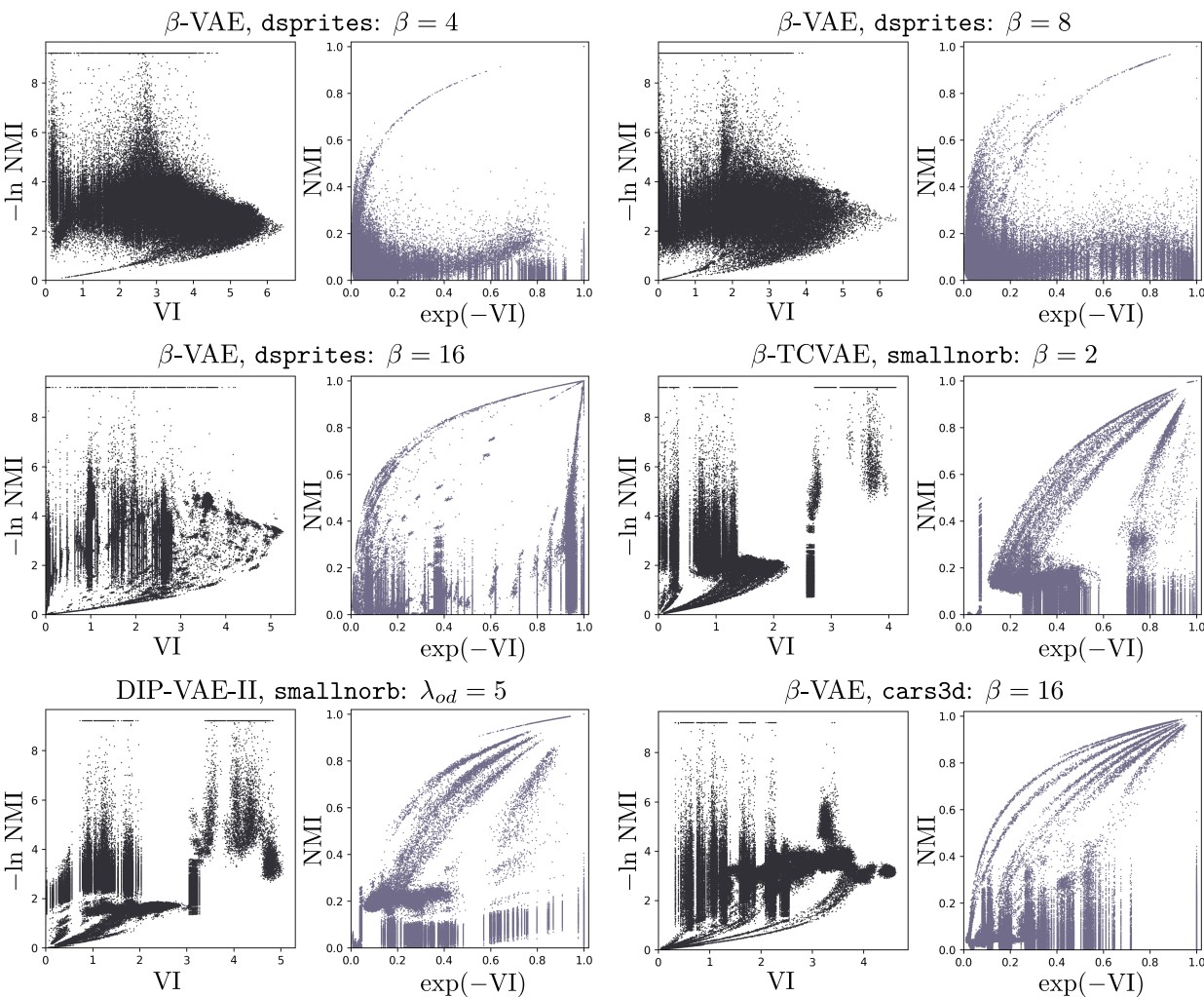

Figure 16: **Direct comparison of NMI and VI.** We convert both measures to a distance measure (black) and to a similarity measure (blue gray) and compare them for the pairwise channel comparisons from the ensembles of Fig. 3.

# E  Appendix: Are NMI and VI interchangeable?

NMI and VI, aside from the inversion required to convert from similarity to distance, can both be seen as a normalized mutual information. Are they interchangeable, or do they assess structure differently?

In Fig. 16, we compare NMI and VI (estimated via Bhattacharyya matrices) as similarity or distance measures for the pairwise comparisons between channels used in Fig. 3. Specifically, as measures of similarity, we plot NMI against $\exp(-\mathrm{VI})$, and for distance we plot $-\log(\mathrm{NMI})$ against VI. We find that NMI and VI are non-trivially related, shown clearly by the horizontal and vertical swaths of points where one of the two measures is roughly constant while the other varies considerably. Interestingly, the corresponding NMI and VI comparisons in Fig. 16 show multiple distinct arcs of channel similarity, as well as clear vertical bands where NMI has discerning power and VI does not.

## F   Appendix: Implementation specifics

Code to reproduce the experiments of Sec. 4 can be found at the following repository: https://github.com/murphyka/representation-space-info-comparison. The heart of the codebase is in `utils.py`, containing the Bhattacharyya and Monte Carlo calculations of $I(U;X)$ and the NMI/VI calculations.

All experiments were implemented in TensorFlow and run on a single computer with a 12 GB GeForce RTX 3060 GPU.

**Models:** For the `dsprites`, `smallnorb`, and `cars3d` datasets, we used the trained models that were publicly released by the authors of Locatello et al. (2019). Thus, all of the model and channel numbers recorded above the latent traversals in Fig. 3b correspond to models that can be downloaded from that paper's github page[3]. Simply add the model offset corresponding to the $\beta = 16$ $\beta$-VAE for `cars3d`, 9250 (e.g., for the traversal labeled with model 31 ch 3, download model 9281 and traverse latent dimension 3, 0 indexed).

For the InfoGAN-CR models on `dsprites`, we used the trained models that were uploaded with Lin et al. (2020)[4].

For results on the training progression of `smallnorb`, `celebA`, and `cars3d`, we used the same architecture and training details from Locatello et al. (2019).

For the MNIST and Fashion-MNIST ensembles, we trained 50 $\beta$-VAEs with a 10-dimensional latent space. The encoder had the following architecture:

> Conv2D: 32 4×4 `ReLU` kernels, stride 2, padding 'same'
>
> Conv2D: 64 4×4 `ReLU` kernels, stride 2, padding 'same'
>
> Reshape([-1])
>
> Dense: 256 `ReLU`
>
> Dense: 20.

The decoder had the following architecture:

> Dense: $7 \times 7 \times 32$ `ReLU`
>
> Reshape([7, 7, 32])
>
> Conv2DTranspose: 64 4×4 `ReLU` kernels, stride 2, padding 'same'
>
> Conv2DTranspose: 32 4×4 `ReLU` kernels, stride 2, padding 'same'
>
> Conv2DTranspose: 1 4×4 `ReLU` kernels, stride 1, padding 'same'.

The models were trained for $2 \times 10^5$ steps, with a Bernoulli loss on the pixels, the Adam optimizer with a learning rate of $10^{-4}$, and a batch size of 64.

**Clustering analysis:** We used the OPTICS implementation from `sklearn`[5] with 'precomputed' distance metric and `min_samples`= 20 (and all other parameters their default values). For distance matrices we converted NMI to a distance with $-\log \max(\text{NMI}, 10^{-4})$.

**Ensemble learning:** For the ensemble learning toy problem (Sec. 4.4), we trained 250 simple $\beta$-VAEs ($\beta$=0.03) whose encoder and decoder were each fully connected networks with two layers of 256 `tanh` activation. The input was two-dimensional, the latent space was one-dimensional, and the output was two-dimensional.

---

[3]https://github.com/google-research/disentanglement_lib/tree/master
[4]https://github.com/fjxmlzn/InfoGAN-CR
[5]https://scikit-learn.org/stable/modules/generated/sklearn.cluster.OPTICS.html

The loss was MSE, the optimizer was Adam with learning rate $10^{-3}$, and the batch size was 2048, trained for 3000 steps. Data was sampled anew each batch, uniformly at random from the unit circle.

To perform ensemble learning, we evaluated the Bhattacharyya matrices for 200 evenly spaced points around the unit circle for each model in the ensemble. Then we directly optimized the parameters for 200 posterior distributions (Gaussians with diagonal covariance matrices) in a two-dimensional latent space, so as to maximize the average similarity (NMI, exponentiated negative VI, or mutual information) between the Bhattacharyya matrix for the trainable embeddings and those of the ensemble. We used SGD with a learning rate of 3 for 20,000 iterations, and repeated for 5 trials for each ensemble size.

**Stochastic shape metrics:** We used publicly released code on `github`[6], using the `GaussianStochasticMetric` with $\alpha = 1$ and the parallelized pairwise distances for the timing calculation.

**CKA:** We replaced the dot-product similarity matrices $K$ and $L$ in the Hilbert-Schmidt Independence Criterion (HSIC) with the Bhattacharyya matrices,

$$\text{HSIC}(\text{BC}^{(1)}, \text{BC}^{(2)}) = \frac{1}{(n-1)^2} Tr(\text{BC}^{(1)} \cdot H \cdot \text{BC}^{(2)} \cdot H), \tag{11}$$

and then followed the prescribed normalization in Kornblith et al. (2019).

**Continuity metric:** Falorsi et al. (2018) used the ratio of neighbor distances in representation space to the corresponding distances in data space, with neighbors taken along continuous paths in data space, as the basis for a discrete continuity metric. It indicated whether any ratios were above some multiplicative factor of some percentile value in the distribution, and thus depended on two parameter choices. Esmaeili et al. (2024) removed one of the parameters—the multiplicative factor—yielding a continuous continuity metric that reports the maximum ratio value over the $90^{th}$ percentile value.

The central premise of this work is to respect the nature of representations as probability distributions, so we used the Bhattacharyya distance (i.e., $D_{ij} = -\log \text{BC}_{ij}$) between posteriors instead of Euclidean distances between posterior means in representation space. Other than this modification, we left the continuity metric as in Esmaeili et al. (2024): as the maximum ratio value over the $90^{th}$ percentile value.

---

[6]https://github.com/ahwillia/netrep

