# OpenReview forum: "Comparing the information content of probabilistic representation spaces"
_TMLR — Accepted by TMLR_

### Review · Reviewer_gsLP · 2024-11-20

**Summary Of Contributions:**

The authors propose a two new similarity metrics between representation spaces. In contrast to typical metrics, the proposed metrics are applicable to compare non-deterministic systems, with representation distributions per input, instead of one deterministic representation per input. The metric is based on estimating mutual information and thus generally applicable regardless of representation space characteristics. They apply the metrics to study representations of VAEs and GANs in the context of disentangled representation learning. The metric is differentiable and can thus be used to directly influence similarity.

**Audience:**

Yes

**Broader Impact Concerns:**

I do not have any concerns with respect to ethical implications of this work.

**Claims And Evidence:**

No

**Requested Changes:**

The following points are critical for acceptance:
* Improved presentation of the paper, e.g., by improving figure captions and taking a bit more space to explain the experiments.
* Add discussion on pros/cons of the new metrics over existing ones.
* Add discussion on the limitations of the proposed metrics and how someone would use them for a new problem.

**Strengths And Weaknesses:**

## Strengths
* (S1) There is only one other metric applicable to stochastic representation spaces and the proposed metrics seem to be more flexible in their application.
* (S2) The metrics themselves are well presented and easy to understand.
* (S3) Understanding of neural network representations is an important problem.

## Weaknesses
* (W1) The biggest problem of the paper lies with its presentation. In general, I found significant parts of the manuscript hard to follow. I think the reason is that the authors assume the reader is familiar with disentangled representation learning, but the paper proposes generally usable similarity metrics and thus should be understandable to a general machine learning audience. A few examples of parts that can be improved:
  * Introduction paragraph 3 and 4: unsupervised evaluation of representation disentanglement is introduced as a motivation but remains vague. Giving an example of what disentangled representations are could help.
  * Contribution 2: “fingerprinting each representation space with the distinguishability of a sample from the dataset”. What does distinguishability mean in this context?
  * Figures are sometimes not self-contained and could benefit from extended captions. For example, for Fig 2, what is the interpretation of the heatmaps? This particularly important as the figures and text, in which they are referenced, are not close to each other. Similarly, for Fig3, there is a ton of information, but the reader is left guessing what it could mean.
  * Section 3.3 has no results?
  * Section 4.4: Please explain the problem better.
* (W2) The metrics are “applicable to general probabilistic representation spaces”. Can you give some general guidelines how to use them? It appears the applications in the paper are limited to low dimensional representations.
* (W3) Comparison with existing metrics. Fig2 shows comparisons to prior metrics CKA and Stochastic Shape Metric. In particular, CKA seems find similarity just as well as NMI, given pairwise distances via Bhattacharyya fingerprint. Does it make sense to use NMI over CKA? What are the pros/cons? Similar for Stochastic Shape Metrics, the similarity estimates seem reasonable. And can the dimensionality issue not be resolved by adding some zero dimensions?

Minor:
* The color scheme in Fig2 goes counter to how similarity is presented popular works such as [1,2].

[1] Kornblith et al., 2019. Similarity of Neural Network Representations Revisited
[2] Raghu et al., 2017. SVCCA: Singular Vector Canonical Correlation Analysis for Deep Learning Dynamics and Interpretability

---

> ### Author Response · Authors · 2025-01-23
>
> We thank the Reviewer for their thoughtful comments and useful feedback.  Below we respond in-line to the questions and comments, and reproduce key parts of the revised text.
>
> > (W1) The biggest problem of the paper lies with its presentation... Introduction paragraph 3 and 4: unsupervised evaluation of representation disentanglement is introduced as a motivation but remains vague. Giving an example of what disentangled representations are could help.
>
> We thank the Reviewer for raising this point and have significantly reworked the text around unsupervised disentanglement in the Introduction and Related work.  Irrelevant text about unsupervised disentanglement has been cut, streamlining paragraphs 3 and 4 in the Introduction and incorporating an example:
>
> “We take as a motivating example the task of unsupervised disentanglement, whose goal is to break information about a dataset into useful factors of variation.  **As an example, a representation space might be trained on images of cars so that color, orientation, and model information are separated into different latent dimensions without any supervision about such factors.  When ground truth factors of variation are unavailable for evaluation, as is generally the case for real-world datasets, existing evaluation methods assess the degree of consensus in an ensemble of trained models.  However, the relatedness of representation spaces has failed to account for the probabilistic nature of the representations, reducing posterior distributions to their means and then using point-based comparisons such as correlation.  With a direct comparison of the information content of representation spaces, we stand to improve the characterization of consensus and of unsupervised disentanglement more generally.**”
>
> > Contribution 2: “fingerprinting each representation space with the distinguishability of a sample from the dataset”. What does distinguishability mean in this context?
>
> We meant distinguishability in terms of the statistical distance between posterior distributions, which allows us to quantify how *distinguishable* two data points are to a downstream model.  In order to clarify what we mean by distinguishability, we have included new results in the synthetic latent space comparison (Fig. 2) that show the qualitative relation between the Bhattacharyya coefficient and the ability of a downstream model to reconstruct the original data.
>
> However, we recognize that distinguishability remains vague by this point in the Introduction.  We have chosen to remove the list of contributions from the Introduction and instead focus the text on our primary contribution -- proposing information-theoretic routes to comparing probabilistic representation spaces.
>
> > Figures are sometimes not self-contained and could benefit from extended captions. For example, for Fig 2, what is the interpretation of the heatmaps? This is particularly important as the figures and text, in which they are referenced, are not close to each other. Similarly, for Fig3, there is a ton of information, but the reader is left guessing what it could mean.
>
> We have expanded the text to most of the figure captions and also moved the figures closer to their relevant text.  The heatmaps in Fig. 2 have also been changed to match what is commonly used for similarity in work using centered kernel alignment (CKA).  The caption for Fig. 3 was previously brief; we have expanded the explanation:
>
> “Figure 3: **Assessing the consistency of channel information in ensembles of models.** We used NMI as a similarity measure for OPTICS to detect fragments of information that are consistently stored in individual channels in an ensemble of trained models. **(a)** The channel consistency of models trained on \texttt{dsprites}, for $\beta$-VAE and InfoGAN-CR. The information with respect to generative factors is shown on the left of each similarity matrix. The $\beta=4$ $\beta$-VAE fragmented information inconsistently compared to the other two ensembles.  **(b)** For a $\beta$-VAE ensemble trained on \texttt{cars3d}, the information content of channels was highly consistent, with seven distinct combinations of the three generative factors. Latent traversals for a representative channel from each grouping visualize the information content. **(c)** We compare the information content of the representatives from panel **b** to that of channels in $\beta$-TCVAE and FactorVAE ensembles. **(d,e)** Channel similarity and latent traversals for $\beta$-VAE ensembles trained on \texttt{fashion-mnist} and \texttt{celebA}.  Additional channel similarity analyses and latent traversals can be found in Appx. A.”

---

> > ### Author Response · Authors · 2025-01-23
> >
> > > Section 3.3 has no results?
> >
> > The results corresponding to Sec. 3.3 (about recovering consistent information fragments) are presented in Sec. 4.2 and Fig. 3, where ensembles of latent spaces are grouped by the OPTICS algorithm.  We have revised the explanation in Sec. 3.3 and reemphasized the connection between the method and results:
> >
> > “Before using OPTICS to group latent dimensions by similarity **(described in Sec. 3.3)**, we removed dimensions transmitting less than 0.01 bits of information.”
> >
> > > Section 4.4: Please explain the problem better.
> >
> > We have revised the text of Section 4.4 and the caption to Fig. 5 for clarity.  The caption to Fig. 5 is now more self-contained as an explanation of the model fusion scenario:
> >
> > "Figure 5: **Fusing weak representation spaces.** **(a)** Example of a one-dimensional latent space of a $\beta$-VAE trained on a dataset generated from a single periodic factor (color hue), which has SO(2) symmetry.  The latent space exhibits flaws where similar values of the generative factor are mapped to dissimilar representations, as seen in the posterior distributions (left) and the distinguishability matrix of \bh coefficients between posteriors, $\text{BC}_{ij}$ (right).  **(b)** We optimized a synthesis representation space to maximize similarity with an ensemble of such one-dimensional latent spaces.  The continuity of statistical distances between neighboring points, an assessment of the fidelity of the global structure of the generative factor, improved as the ensemble size grew.  Error bars show the standard deviation over five experiments, and values are offset horizontally for visibility.  **(c)** Synthesized two-dimensional representation spaces (posterior means shown as points; covariances as shaded ellipses) and their corresponding distinguishability matrices. Panels compare results when maximizing average NMI (left, middle) and mutual information (right)."
> >
> > > (W2) The metrics are “applicable to general probabilistic representation spaces”. Can you give some general guidelines on how to use them? It appears the applications in the paper are limited to low dimensional representations.
> >
> > While the proposed measures’ use of mutual information grants flexibility, estimation can indeed be more difficult to assess in higher dimensional spaces.  We have added the following sentence to the Discussion:
> >
> > “While the examples in this work focused on relatively low-dimensional latent spaces that are common in practice, scaling to higher-dimensional representation spaces may face challenges related to the reliable estimation of mutual information (McAllester & Stratos, 2020).”
> >
> > > (W3) Comparison with existing metrics. Fig2 shows comparisons to prior metrics CKA and Stochastic Shape Metric. In particular, CKA seems to find similarity just as well as NMI, given pairwise distances via Bhattacharyya fingerprint. Does it make sense to use NMI over CKA? What are the pros/cons?
> >
> > Thank you for raising this important point. The form of CKA shown in Fig. 2, where pairwise similarities of representations are computed via Bhattacharyya coefficients, indeed is highly similar to NMI when applied to probabilistic representations. However, it is important to note that this variation of CKA is an ad hoc adaptation, and to our knowledge, it has not been introduced or theoretically justified in prior work.  By contrast, the NMI formulation used in our method is grounded in information theory and explicitly designed to compare the information content of probabilistic representation spaces.
> >
> > We have revised the discussion in Sec. 4.1 to better highlight the distinctions between these approaches and their respective advantages.  Additionally, we have added two variants of CKA that are more standard in the literature: CKA-linear---the most commonly used form of CKA--- and CKA-nonlinear---meant to capture topological similarity as opposed to strict geometrical similarity---and applied both to the means of the posterior distributions.  We think that the inclusion of what people commonly refer to as CKA helps disambiguate the relation between NMI and the version of CKA based on Bhattacharyya similarity that we introduced.
> >
> > Relevant addition to Sec. 4.1: “NMI relates the latent spaces in much the same way as the Bhattacharyya variant of CKA. While this *ad hoc* variant of CKA lacks the information-theoretic underpinnings of NMI, it offers an easy-to-use alternative by simply replacing a point-based distance with a statistical distance between representations.”

---

> > > ### Author Response · Authors · 2025-01-23
> > >
> > > > Similar for Stochastic Shape Metrics, the similarity estimates seem reasonable. And can the dimensionality issue not be resolved by adding some zero dimensions?
> > >
> > > We appreciate the observation that similarity estimates for Stochastic Shape Metrics are reasonable, particularly under the premise that certain transformations (such as rotations) are considered equivalent when comparing representation spaces. However, from the perspective of downstream processing, we believe that information-theoretic measures more directly capture the sense of similarity we aim to evaluate. To clarify this point, we have added new results to Fig. 2, showing the Spearman correlation between the similarity estimates of downstream classification predictions and various methods for comparing latent spaces.
> > >
> > > Regarding the dimensionality issue, we agree that adding zero dimensions can address this for point-based shape metrics, as demonstrated in Williams et al. (2021). For the stochastic extension to shape metrics, however, we did not find guidance in the paper or its codebase on how zero dimensions might be incorporated before optimizing over transformations and computing the Wasserstein distance.  While this may be technically feasible, addressing such modifications to their method is beyond the scope of our work.
> > >
> > > >The color scheme in Fig2 goes counter to how similarity is presented in popular works such as [1,2].
> > >
> > > Fixed the color scheme, thank you for pointing this out.

---

> > > > ### Comment · Reviewer_gsLP · 2025-01-29
> > > >
> > > > Thank you for all the clarifications. All my concerns have been addressed and I think the paper is in a good state now.
> > > >
> > > > Just one minor suggestion that could improve the paper marginally: In Section 4.2., mentioning that the ensemble consists of 50 models with 10 dimensions each could be helpful. You already reference the section where this is mentioned, but it felt a bit unclear when rereading.

---

> > > > > ### Author Response · Authors · 2025-01-30
> > > > >
> > > > > Good suggestion, thank you.  We added this information to the beginning of Sec. 4.2:
> > > > >
> > > > > "We next analyzed the consistency of information fragmentation in ensembles of generative models trained on image datasets. **Using fifty models with ten latent dimensions (channels) each, for a variety of datasets, methods, and hyperparameters (some of which were released with Locatello et al. (2019)), we assessed structure in an ensemble's channels using the proposed similarity measures.**"

---

### Review · Reviewer_ouLB · 2024-11-28

**Summary Of Contributions:**

Quantifying geometric similarity in neural representations is an active topic of research in machine learning (see e.g. [Klabunde et al.](https://arxiv.org/abs/2305.06329) for a recent review). This paper proposes a novel approach to quantify similarity in the information content of probabilistic representation spaces, which occur (for example) in the bottleneck layer of variational autoencoder. The authors claim the method will be helpful in evaluating the level of disentanglement in learned representations, which is also an important and challenging topic (see e.g. [Locatello et al.](https://proceedings.mlr.press/v97/locatello19a.html)).

**Audience:**

Yes

**Claims And Evidence:**

No

**Requested Changes:**

I covered major points under weaknesses above. In general, I don't have any strong feelings against this paper. I would just like to see an honest effort made at clarifying the points listed above.

**Strengths And Weaknesses:**

Strengths:

* Few papers deal with comparing probabilistic or stochastic representations. One of the only other examples is the recent work of [Duong et al.](https://arxiv.org/abs/2211.11665). Thus, the paper is addressing a key gap in the literature.

* The paper leverages information theoretic concepts, which is also novel in this area as far as I am aware.

Weaknesses:

* While I like creative ideas in this paper, the exposition is confusing and I am ultimately unsure of the motivation behind the method. I'll expand on this in the points below.

* One point of confusion is the notion that probabilistic representations can be thought of as "soft clusterings" of datapoints. What the authors mean by this isn't really spelled out in the paper, and to me this doesn't comport with the usual meaning of "soft clusters." My understanding of k-means clustering (see e.g. [Zhu et al.](https://arxiv.org/abs/2403.15700)) typically involves assigning each data point, $\mathbf{x}_i$ for $i \in \{1, \dots, n\}$, to a probability vector $\mathbf{p}_i$ that takes values on a the $k$-simplex (for k clusters). The natural probabilistic representation that corresponds to this would be a random variable following a Dirichlet distribution, but typically the latent space of a VAE is defined by an isotropic Gaussian prior. Interestingly, some papers have been published on Dirichlet VAEs (such as this paper by [Joo et al.](https://arxiv.org/abs/1901.02739)). Here, I can see how the latent space would correspond to a soft clustering, and it could be a very useful way for the authors to apply their method!

* A similar point of confusion for me is why clustering is a good framework to ground comparisons of neural representations. The authors say at the beginning of section 3 that "Our goal is to compare the information transmitted about a dataset by different representation space" but then later say that the information transmitted for hard clustering is simply the entropy of the cluster labels. Thus, the optimal representation for $n$ datapoints is just a representation that puts every datapoint into a singleton cluster, which doesn't seem like a particularly useful representation. I suspect that I'm missing the point, but I find it hard to understand the rest of the paper unless the authors flesh out their reasoning here.

* The Monte Carlo estimator of mutual information seems critical, but is only described briefly in appendix E. I feel pretty strongly that it should be put into the main text / methods section.

* I expect that the Monte Carlo estimator will perform poorly for high dimensional latent spaces due to standard curse of dimensionality arguments. I gather that this is why the authors look at relatively low-dimensional latent spaces in their experiments (up to ten dimensions are considered for Fig 4). The authors should be more up front about these limitations and discuss the statistical estimation properties (e.g. how are the error bars in Fig 4 computed?).

* Moreover, the Monte Carlo estimator only gives an estimate for the mutual information between the empirical distribution over $\mathbf{x}$ and the latent representation. I would be interested in a plot that shows whether this estimator is converging to something reasonable as a function of the dataset size (i.e. number of images). In particular if you take random 50-50 splits of the data and estimate the mutual information on each subset, how much disagreement is there?

* In figure 2, I am not sure whether the proposed method is directly comparable to the baselines the authors compare to: CKA and stochastic shape distances. Neither of these baselines adopt the clustering perspective (either hard or soft clustering), so I don't know how I'm supposed to compare with NMI? Perhaps the authors could re-word this section more neutrally. For example, the shape metric and (to a lesser extent) CKA results in Fig 2 show that these methods do a good job of separating out representations into three clusters based on their geometry---i.e. they are behaving as they were intended to behave. The authors seem to prefer their approach, which is a "more flexible sense of [representational] similarity". By this, I guess that they want a method that is more sensitive to the topology (rather than geometry) of the probabilistic point cloud? If that is the case, I think it could help to state that and explain why certain circumstances might call for a geometric vs topological measure. Furthermore, the authors should discuss nonlinear CKA and topological RSA, which may be more fair baselines to compare against (see the papers by [Lin & Kriegeskorte](https://doi.org/10.1073/pnas.2317881121) and [Williams, 2024](https://www.biorxiv.org/content/10.1101/2024.10.23.619871v1.abstract))

---

> ### Author Response · Authors · 2025-01-23
>
> We thank the Reviewer for their close reading of our paper and the constructive feedback, and respond in-line to the raised points below.
>
> > One point of confusion is the notion that probabilistic representations can be thought of as "soft clusterings" of datapoints. What the authors mean by this isn't really spelled out in the paper, and to me this doesn't comport with the usual meaning of "soft clusters."
>
> Thank you for pointing out this potential source of confusion.  The common usage of soft clustering typically maps data points to a finite number of clusters, whereas our work builds upon a broader notion where data points are mapped to a continuum.  In both cases, the membership for each datum sums to one and is equivalent to a probability distribution.  We have clarified this connection in the Related Work and Methods sections as follows:
>
> “We observe that a probabilistic representation space communicates for each datum a soft assignment over latent vectors, with the degree of membership expressed by the probability density of posterior distributions.” (Related Work) and “While soft clustering is predominantly performed over a discrete set of clusters, here we view each point $u$ in a continuous latent space as a cluster, with the posterior distribution $p(u|x)$ assigning membership over the continuum.” (Methods)
>
> This connection with clustering serves primarily as an intuitive motivation for extending information-theoretic measures like NMI and VI to probabilistic representation spaces.  Once these generalized measures are derived, the connection with clustering becomes less central to the method's use and interpretation.
>
> > The natural probabilistic representation that corresponds to this would be a random variable following a Dirichlet distribution, but typically the latent space of a VAE is defined by an isotropic Gaussian prior. Interestingly, some papers have been published on Dirichlet VAEs (such as this paper by Joo et al.). Here, I can see how the latent space would correspond to a soft clustering, and it could be a very useful way for the authors to apply their method!
>
> We agree that latent spaces representing probability distributions over a discrete set of outcomes are closely aligned with typical notions of soft clustering. Exploring such spaces, for example through methods like Dirichlet VAEs, could indeed be an interesting direction for further investigation. Thank you for the thoughtful suggestion!
>
> > A similar point of confusion for me is why clustering is a good framework to ground comparisons of neural representations. The authors say at the beginning of section 3 that "Our goal is to compare the information transmitted about a dataset by different representation space" but then later say that the information transmitted for hard clustering is simply the entropy of the cluster labels. Thus, the optimal representation for 𝑛 datapoints is just a representation that puts every datapoint into a singleton cluster, which doesn't seem like a particularly useful representation. I suspect that I'm missing the point, but I find it hard to understand the rest of the paper unless the authors flesh out their reasoning here.
>
> The reviewer raises an important point. While placing each datapoint into a singleton cluster would indeed maximize transmitted information, this alone does not make for a useful representation. For example, a representation space trained with SimCLR (Chen et al., 2020) might preserve semantic information from natural images while discarding nuisance variation, such as pixel jitter. Its utility lies equally in what it retains and what it removes. We have added the following sentences to the Methods:
>
> “We note that maximizing communicated information, such as by assigning each data point to its own cluster, does not yield a useful representation.  The value of a representation lies in the balance between the information preserved and the irrelevant variation discarded, highlighting the importance of assessing its specific information content.“
>
> > The Monte Carlo estimator of mutual information seems critical, but is only described briefly in appendix E. I feel pretty strongly that it should be put into the main text / methods section.
>
> Thank you for the suggestion.  We have added the following description of the Monte Carlo estimator in Sec. 3.1 (Methods), and added more detail about it in Appendix C.
>
> “We propose two routes to estimating NMI and VI that offer a tradeoff between precision and speed; both leverage the known posterior distributions to calculate the information transmitted about the dataset by combinations of representation spaces, $I(X;\cdot)$.
> The first route is to compute $I(X;\cdot)$ with a straightforward Monte Carlo estimate using the aggregated posterior over the entire dataset of size $L$,
>
> $I(X;U) = E_{x \sim p(x)} E_{u \sim p(u|x)} \left [ \log \frac{p(u|x)}{\sum_i^L p(u|x_i)} \right ].$
> “

---

> > ### Author Response · Authors · 2025-01-23
> >
> > > I expect that the Monte Carlo estimator will perform poorly for high dimensional latent spaces due to standard curse of dimensionality arguments. I gather that this is why the authors look at relatively low-dimensional latent spaces in their experiments (up to ten dimensions are considered for Fig 4). The authors should be more up front about these limitations and discuss the statistical estimation properties (e.g. how are the error bars in Fig 4 computed?).
> >
> > We focused on low-dimensional latent spaces because they are used in practice for the datasets considered, and simply used a number of samples for the MC estimator such that mutual information uncertainty was on the order of 0.01 bits (taken as the standard error, i.e. the standard deviation divided by the square root of the number of samples).  To achieve similar uncertainty for higher-dimensional spaces might require more MC samples, but should not present a fundamental barrier to analysis.  That said, we have added the following sentence to the Discussion:
> >
> > “While the examples in this work focused on relatively low-dimensional latent spaces that are common in practice, scaling to higher-dimensional representation spaces may face challenges related to the reliable estimation of mutual information (McAllester & Stratos, 2020).”
> >
> > The error bars on NMI and VI were obtained through uncertainty propagation.  We have added a more detailed explanation to Appendix C (“Information estimation using Bhattacharyya distinguishability matrices and Monte Carlo estimator”), with the portion of primary relevance below:
> >
> > “The uncertainty was calculated by propagating the uncertainty of Monte Carlo estimates of $I(X;U)$ according to the expression for VI or NMI.
> > To be specific, as $\text{VI}(U,V)=2I(X;U,V)-I(X;U,U^\prime)-I(X;V,V^\prime)$, the propagated uncertainty is given by
> > \begin{equation}
> >     \Delta \text{VI}(U,V)
> >     =\sqrt{4 \Delta I(X;U,V)^2 - \Delta I(X;U,U^\prime)^2  - \Delta I(X;V,V^\prime)^2},
> > \end{equation}
> > with $\Delta I(X;U,V)$ the standard error from the Monte Carlo estimate.
> > The expression for the uncertainty on the NMI estimate contains many more terms, and we will not reproduce them here.
> > Finally, the error bars in Fig. 4 are the standard error of the weighted mean,
> > \begin{equation}
> >     \Delta Q = \left ( \sum_i^n (\Delta Q_i)^{-2} \right )^{-\frac{1}{2}},
> > \end{equation}
> > for quantity $Q$.”
> >
> > > Moreover, the Monte Carlo estimator only gives an estimate for the mutual information between the empirical distribution over 𝑥 and the latent representation. I would be interested in a plot that shows whether this estimator is converging to something reasonable as a function of the dataset size (i.e. number of images). In particular if you take random 50-50 splits of the data and estimate the mutual information on each subset, how much disagreement is there?
> >
> > This is a valid point, relevant for any information theoretic analysis of a representation space: that quantities will generally be with respect to the empirical distribution rather than to the ground truth generating distribution.  We evaluate with the full dataset so that the evaluation’s empirical distribution matches what was used for training.  However, due to the computational costs involved, one might wonder if it is necessary to use the full training dataset.  We evaluated the Monte Carlo estimator's stability by sampling random subsets of varying sizes of the dataset and repeating this process several times for each size.  The results, shown in Fig. 14 and discussed in Appx. C, exhibit systematic underestimation as less of the data is used to compute the aggregated posterior, though the effect is relatively small (O(0.1) bits when using 10% of the data).

---

> > > ### Author Response · Authors · 2025-01-23
> > >
> > > > In figure 2, I am not sure whether the proposed method is directly comparable to the baselines the authors compare to: CKA and stochastic shape distances. Neither of these baselines adopt the clustering perspective (either hard or soft clustering), so I don't know how I'm supposed to compare with NMI? Perhaps the authors could re-word this section more neutrally. For example, the shape metric and (to a lesser extent) CKA results in Fig 2 show that these methods do a good job of separating out representations into three clusters based on their geometry---i.e. they are behaving as they were intended to behave. The authors seem to prefer their approach, which is a "more flexible sense of [representational] similarity".
> > >
> > > We recognize that the original presentation of Fig. 2 was overly qualitative and poorly explained.  We have revamped Fig. 2 to include quantitative comparisons between the methods (in the form of Spearman’s rank correlation), a point of reference based on the information available for downstream processing (the Jensen-Shannon divergence between classification head outputs), and greatly expanded the discussion in Sec. 4.1.
> > >
> > > We emphasize that the clustering perspective is only used to motivate our methods of comparison, which were first proposed in literature on hard clustering.  There is no clustering perspective needed to interpret Fig. 2: there are simply different methods with which to compare the nine synthesized latent spaces.
> > >
> > > > By this, I guess that they want a method that is more sensitive to the topology (rather than geometry) of the probabilistic point cloud? If that is the case, I think it could help to state that and explain why certain circumstances might call for a geometric vs topological measure. Furthermore, the authors should discuss nonlinear CKA and topological RSA, which may be more fair baselines to compare against (see the papers by Lin & Kriegeskorte and Williams, 2024)
> > >
> > > We thank the reviewer for pointing us to recent literature focusing on topology rather than geometry, and have included nonlinear CKA in our comparison of synthesized representation spaces (Sec. 4.1, Fig. 2).  The cited works’ recognition of topology as more relevant than geometry for further processing aligns with our motivation: we wish to assess the content that would be available to an expressive network downstream, which should not depend on the precise manner by which distant representations are arranged.  If representations are probabilistic, information theory can be used according to our method **without any hyperparameters**, assessing information content directly.  On the other hand, if representations are point-based, then topological RSA or nonlinear CKA can be used with hyperparameters that set distance scale(s) in representation space.
> > >
> > > We have added the following text to the Related Work:
> > >
> > > “A rich area of research compares point-based representation spaces via the pairwise geometric similarity of a common set of data points in the space (Kornblith et al., 2019; Hermann & Lampinen, 2020; Klabunde et al., 2023), building upon representational similarity analysis from neuroscience (Kriegeskorte et al., 2008).  **In place of geometric similarity, topological similarity more closely probes the information available for downstream processing by employing specific similarity functions with tunable parameters (Lin & Kriegeskorte, 2024; Williams, 2024).**  For comparing probabilistic representation spaces, stochastic shape metrics (Duong et al., 2023) extend a distance metric between point-based representation spaces based on aligning one with another through prescribed transformations (e.g., rotations)  (Williams et al., 2021). **By contrast, the information-theoretic lens we adopt requires no enumeration of transformations nor tuning of parameters, and directly assesses the information available for processing by downstream neural networks.**“

---

### Review · Reviewer_1uJU · 2024-12-23

**Summary Of Contributions:**

This paper investigates the problem of comparing the information content of probabilistic representation spaces and introduces a generalization of two information-theoretic methods for such comparisons. The authors propose a novel, lightweight approach based on fingerprinting the representation space using the Bhattacharyya coefficient. The paper further demonstrates the applicability of this method through a series of experiments, highlighting its potential for model fusion and unsupervised detection of model structures.

**Audience:**

Yes

**Claims And Evidence:**

No

**Requested Changes:**

**Critical Requested Changes:**
- Address the issues raised in the "Weaknesses" section.

**Would Strengthen the Work:**
1. **Figure 3** – Additional description and details would improve clarity regarding the OPTICS reachability profile and the clustering process. Expanding on how the clustering was performed would enhance the reader’s understanding.
2. **Experiment 4.1 and Figure Placement** – Experiment 4.1 (page 8) references Figure 2 (page 6) for the first time, which disrupts the reading flow. Consider adjusting the figure placement to improve readability.
3. **Applications to Representational Alignment** – Discussing potential applications of the proposed method for representational alignment could add further value to the paper [2].

[2] Muttenthaler, Lukas, et al. "Aligning Machine and Human Visual Representations across Abstraction Levels." arXiv preprint arXiv:2409.06509 (2024).

**Strengths And Weaknesses:**

### Strengths

The paper is clearly written and generally easy to follow.  The intuition behind the proposed approach is well-articulated, and the authors provide supporting evidence for most of their claims. The paper explores a diverse set of datasets, including a balanced combination of real-world and synthetic data, to evaluate the properties of the proposed method. Results on unsupervised detection of structure and model fusion look very promising and interesting.

### Weaknesses
While I remain positive about the paper, one critical weakness, in my view, is the limited amount of quantitative evidence and comparisons to existing methods — particularly regarding the claims that the proposed approach is *fast* and *lightweight*.

- For instance, while the paper claims that the proposed measures are *lightweight* (p.2) and *fast* (p.5), the evidence for these assertions is primarily qualitative. Experiment 4.1 estimates runtimes for individual methods, but the supporting data lacks quantitative rigor.
- Additionally, on page 8, the authors state that "NMI and VI are nearly the same as the Monte Carlo alternatives". However, this claim is assessed only through visual comparison of matrices in Figure 2.

While I find the proposed method promising, I recommend a more rigorous and quantitative demonstration of the following:
1. **Runtime Comparison** – Vary key parameters and quantitatively show that the proposed method consistently outperforms alternatives in terms of speed.
2. **Result Similarity** – Compare the proposed method with other approaches, such as CKA, and provide quantitative evaluations (e.g., norm differences between similarity matrices?).

**Question to the Authors:**
While it is understandable that evaluating methods in unsupervised settings can be challenging, a potential solution could be to demonstrate the stated claims in a supervised setting. By using a scenario where some ground truth is available, the performance of various methods could be quantitatively assessed on this (potentially toy) example —(for example, [1]), or in the setting similar to the Experiment 4.1. Would the authors consider this as a viable path to strengthen the empirical validation of the proposed method?

[1] Bykov, Kirill, et al. "DORA: Exploring Outlier Representations in Deep Neural Networks." Transactions on Machine Learning Research.

---

> ### Author Response · Authors · 2025-01-23
>
> We thank the Reviewer for their close reading of our work and the constructive criticism.  We have bolstered the paper with quantitative evidence and additional exposition.  Below we respond to specific points.
>
> > While I remain positive about the paper, one critical weakness, in my view, is the limited amount of quantitative evidence and comparisons to existing methods — particularly regarding the claims that the proposed approach is *fast* and *lightweight*... Runtime Comparison – Vary key parameters and quantitatively show that the proposed method consistently outperforms alternatives in terms of speed.
>
> We appreciate the reviewer’s feedback regarding the claims of speed and computational feasibility. To clarify, our intent was not to position the proposed method as superior in runtime compared to alternatives, but rather to emphasize its computational practicality in addressing the comparison of probabilistic representation spaces. We have revised the text so that efficiency is presented as a practical feature rather than a central axis of comparison.
>
> To address the reviewer’s concern about runtime evidence, we highlight runtime comparisons provided for the synthetic example in Sec. 4.1, where our method was approximately two orders of magnitude faster than the most relevant baseline, stochastic shape metrics (Duong et al., 2023).  Additionally, the practicality of evaluating all pairwise channel similarities across 500 channels (Sec. 4.2) and performing gradient descent for model fusion (Sec. 4.4) relied on the computational efficiency of our Bhattacharyya coefficient-based estimators for NMI and VI.  We believe these examples demonstrate the scalability and feasibility of the proposed method in practical scenarios, but we welcome further suggestions on additional quantitative evidence that could strengthen this aspect of the work.
> > Additionally, on page 8, the authors state that "NMI and VI are nearly the same as the Monte Carlo alternatives". However, this claim is assessed only through visual comparison of matrices in Figure 2... *Result Similarity* – Compare the proposed method with other approaches, such as CKA, and provide quantitative evaluations (e.g., norm differences between similarity matrices?).
>
> We appreciate this suggestion, and have included the Spearman rank correlation of latent space similarities across different measures in Fig. 2 as a quantitative evaluation.  The lightweight Bhattacharyya approximation and the Monte Carlo estimates are highly correlated.  VI also correlates highly with the similarity of outputs of a classification head (please see the response following this one), while NMI correlates with our ad hoc variant of CKA that uses a statistical distance as its basis of comparison.
>
> > While it is understandable that evaluating methods in unsupervised settings can be challenging, a potential solution could be to demonstrate the stated claims in a supervised setting. By using a scenario where some ground truth is available, the performance of various methods could be quantitatively assessed on this (potentially toy) example —(for example, [1]), or in the setting similar to the Experiment 4.1. Would the authors consider this as a viable path to strengthen the empirical validation of the proposed method?
>
> This was another helpful suggestion for grounding the results of Fig. 2 and building intuition around the proposed similarity measures.  Our central goal is to assess the information available for downstream processing, and we can view the performance of a downstream classification network as a sort of ground truth.  In the revised Sec. 4.1 / Fig. 2, we have trained a classification head on top of each synthesized latent space to predict the input from a sample from the latent space, and used the similarity of output predictions (quantified by Jensen-Shannon divergence) as a point of reference for the various latent space similarity measures.  We find that VI closely correlates with the similarity of outputs by the downstream models, and also that the Bhattacharyya fingerprint of the latent space captures nearly the same structure of confusion between data points as the predictions of the classification head.

---

> > ### Author Response · Authors · 2025-01-23
> >
> > > Figure 3 – Additional description and details would improve clarity regarding the OPTICS reachability profile and the clustering process. Expanding on how the clustering was performed would enhance the reader’s understanding.
> >
> > We have revised the caption of Fig. 3, the text of Sec. 3.2, and more clearly linked this part of the paper to the discussion of experimental results in Sec. 4.2.  The expanded caption of Fig. 3:
> >
> > "Figure 3: **Assessing the consistency of channel information in ensembles of models.**  We used NMI as a similarity measure for OPTICS to detect fragments of information that are consistently stored in individual channels in an ensemble of trained models. **(a)** The channel consistency of models trained on \texttt{dsprites}, for $\beta$-VAE and InfoGAN-CR.  The information with respect to generative factors is shown on the left of each similarity matrix. The $\beta=4$ $\beta$-VAE fragmented information inconsistently compared to the other two ensembles.  **(b)** For a $\beta$-VAE ensemble trained on \texttt{cars3d}, the information content of channels was highly consistent, with seven distinct combinations of the three generative factors.  Latent traversals for a representative channel from each grouping visualize the information content. **(c)** We compare the information content of the representatives from panel **b** to that of channels in $\beta$-TCVAE and FactorVAE ensembles. **(d,e)** Channel similarity and latent traversals for $\beta$-VAE ensembles trained on \texttt{fashion-mnist} and \texttt{celebA}.  Additional channel similarity analyses and latent traversals can be found in Appx. A.”
> >
> > > Experiment 4.1 and Figure Placement – Experiment 4.1 (page 8) references Figure 2 (page 6) for the first time, which disrupts the reading flow. Consider adjusting the figure placement to improve readability.
> >
> > We have repositioned all figures more closely to their corresponding text.
> >
> > > Applications to Representational Alignment – Discussing potential applications of the proposed method for representational alignment could add further value to the paper.
> >
> > Thank you for the suggested connection.  We have added to the Discussion:
> >
> > “With differentiable formulations for NMI and VI, model fusion and the more general problem of representational alignment (Sucholutsky et al., 2023; Muttenthaler et al., 2024) can be effectively approached from an information theoretic perspective. Potential applications include aligning representation spaces across models trained on different subsets of data, improving ensemble methods, and evaluating consistency of representations in multitask learning or domain adaptation, where reconciling heterogeneous latent spaces is often crucial.“

---

### Author Response · Authors · 2025-01-23

We appreciate the valuable feedback from all the reviewers as it helped us strengthen the paper.  We are happy that the paper was seen as “**clearly written**” with “**well-articulated [intuition]**” (1uJU) and “**very promising and interesting [results]**” (1uJU), addressing “**an important problem**” (gsLP) / “**a key gap in the literature**” (ouLB) with a “**novel approach**” containing “**creative ideas**” (ouLB).
In response to the reviewers’ constructive critiques, we have addressed concerns related to the overall motivation and its connection to soft clustering, the qualitative nature of the synthetic example (Sec. 4.1, Fig. 2), and the explanation of the Monte Carlo estimator used in our method. Key revisions include:
- **Additional Results**: New experiments and an updated discussion in Sec. 4.1 (Fig. 2) to provide more robust comparisons of similarity measures.
- **Improved Framing**: Clarified the framing of our method in terms of soft clustering and its relevance to unsupervised disentanglement (revised Introduction, Related Work, and Methods sections).
- **Expanded Explanation**: Detailed the Monte Carlo estimation approach and the uncertainty propagation used to derive the error bars in Fig. 4.

We have uploaded a marked-up PDF ([here](https://github.com/comparingTMLR/comparingTMLR/blob/main/Comparing%20the%20information%20content%20of%20probabilistic%20representation%20spaces.pdf)) highlighting the changes, and have listed relevant edits in our detailed responses below.

---

### Decision · Action_Editor_yK3D · 2025-02-13

**Recommendation:** Accept as is

**Comment:**

After the authors' revisions, the paper has improved substantially. All three reviewers think that their main concerns have been addressed and that the paper meets the acceptance criteria.

**Audience:**

The paper is overall sound and of interest to the TMLR audience.

**Claims And Evidence:**

This paper proposes two information-theoretic measures to compare probabilistic representation spaces, extending methods used for comparing hard clustering assignments. These measures address the limitations of existing methods that often overlook the distributional
nature of probabilistic representation spaces. The paper also introduces a fast estimation method based on fingerprinting a representation space with a data sample, which is particularly useful when information is limited to a few bits.

The paper is well written, technically correct, and identifies a clear gap in the literature. Reviewer ouLB acknowledges that there are some oustanding questions that could be reasonably deferred to future work. Reviewer 1uJU finds that two of the main claims of the paper are somewhat supported by evidence, namely: (1) that the method is lightweight and fast (this is demonstrated with an experiment), and (2) that the quantitative evaluations are adequate (which was addressed by the authors by including a new analysis using Spearman rank correlation).

Despite the fact that further experiments would definitely improve the paper (e.g., Reviewer gsLP points out that current results are for toy problems and low-dimensional representations only), all three reviewers think that the paper is above the acceptance bar, or at least "leaning towards acceptance".